# COGNAC: Cooperative Graph-based Networked Agent Challenges for Multi-Agent Reinforcement Learning

**Jules Sintes**

Inria and DI ENS, École Normale Supérieure, PSL University Paris, France

**Ana Bušić**

Inria and DI ENS, École Normale Supérieure, PSL University Paris, France

## Abstract

Many controlled complex systems have an inherent network structure, such as power grids, traffic light systems, or computer networks. Automatically controlling these systems is highly challenging due to their combinatorial complexity. Standard single-agent reinforcement learning (RL) approaches often struggle with the curse of dimensionality in such settings. In contrast, the multi-agent paradigm offers a promising solution by distributing decision-making, thereby addressing both algorithmic and combinatorial challenges. In this paper, we introduce COGNAC (COoperative Graph-based Networked Agent Challenges), a collection of cooperative graph-structured environments designed to facilitate experiments across different graph sizes and topologies. COGNAC bridges the gap between theoretical research in network control and practical multi-agent RL (MARL) applications by offering a flexible, scalable platform with a suite of simple yet highly challenging problems rooted in networked environments. Our benchmarks also support the development and evaluation of decentralized and distributed learning algorithms, motivated by the growing interest in more sustainable and frugal AI systems. Experiments on COGNAC show that independent actor–critic learning (IPPO) yields the highest-quality joint policies while scaling robustly to large network sizes with minimal hyperparameter tuning. Value-based independent learning (IDQL) typically needs substantially more training and is less reliable on combinatorial tasks. In contrast, standard Centralized-Training Decentralized-Execution (CTDE) methods and fully centralized training are slower to converge, less stable, and struggle to generalize to larger, more interdependent networks. These results suggest that CTDE approaches likely need extra information or inter-agent communication to fully capture the underlying network structure of each problem.

## 1   Introduction

Many real-world systems exhibit an inherent graph structure or can be naturally modeled as networks. These systems can be found in a wide variety of applications and theoretical fields: computer networks, biological networks, social networks, power grids, or logistic networks. Graph-based problems are widely studied from a theoretical point of view in the context of graph theory. However, applying optimization and machine learning methods to networked real-life systems is often highly difficult. Indeed, the size of networks and the complexity of control problems tend to make such problems intractable for standard optimization or centralized machine learning algorithms. In this paper, we deliver an intermediate platform with a collection of fully collaborative multi-agent environments

39th Conference on Neural Information Processing Systems (NeurIPS 2025) Track on Datasets and Benchmarks.

with inherent network structure dedicated to reinforcement learning (RL). This work aims to foster the development of scalable and frugal decentralized methods by introducing minimal yet challenging benchmark environments tailored for multi-agent systems with a networked structure.

Multi-agent reinforcement learning (MARL) has emerged as a powerful framework for enabling autonomous agents to make sequential decisions in complex, dynamic environments. At the core of MARL lies the classical Markov Decision Process (MDP) framework, which provides a rigorous mathematical basis for modeling decision-making in stochastic settings and optimizing long-term cumulative rewards [1]. However, in many real-world scenarios, agents operate under conditions of uncertainty and partial observability, necessitating the use of Partially Observable MDPs (POMDPs) to capture incomplete and noisy state information [2]. Extending these frameworks to multi-agent systems brings new challenges like coordination, decentralized information, and non-stationarity. This has led to growing interest in decentralized POMDPs as a foundation for modeling cooperative and competitive interactions [3].

## 1.1 Distributed Reinforcement Learning with Graph Structure

Reinforcement learning (RL) has proven to be highly efficient for many applications, such as robotics, autonomous vehicles, video games, etc. Concurrently, the emergence of deep learning over the last decade has allowed the development of model-free RL algorithms for many highly complex tasks. Still, complex network systems control, that could use RL methods, struggle with the curse of dimensionality. Hence, while standard centralized RL often works on small instances of problems, the scalability of methods to larger systems, closer to the real ones, is often impossible [4].

The main challenge in tackling DecPOMDP problems lies in the non-stationarity induced by the presence of multiple agents that partially observe the system and act simultaneously [3]. Coordinated collaborative MARL was first introduced as a paradigm aiming to leverage the inherent graph structure of multi-agent problems to find an optimal joint policy. Initially, the main interest in decentralized methods was to avoid exploring the complete joint state-action space, as it is intractable even for small instances of simple problems [5]. Through the learning of a structured representation of the policy or value function, leveraging directly the relationships between agents, it becomes possible to approximate optimal policies. This is particularly useful in the case of Factored MDPs [6] and Network-Distributed MDPs [7]. More recent work on coordinated multi-agent systems has introduced methods to tackle the exponentially growing state-action space by exploiting the network structure of problems to learn policies locally [8, 9].

More generally, the recent growing interest in distributed machine learning methods in a decentralized paradigm offers many opportunities to tackle control problems for vital infrastructures of increasing size and complexity in the fields of energy, telecommunication, and operational research [10]. Hence, the development of a collection of benchmark environments with inherent network structures is driven by the emergent need for evaluation tools to support the development of such distributed RL methods for complex systems. In the context of control problems, one can leverage these paradigms to transform large intractable instances of a problem into a smaller set of subproblems. In terms of computation, these algorithms may be distributed, making them flexible and frugal solutions for constrained hardware resources. In addition, they may offer more scalability for very large multi-agent systems in the context of partially observable environments with constrained communication [11, 12].

## 1.2 Review of Existing Benchmark Environments

Most of the available RL environments have historically focused on the centralized single-agent paradigm, as it has been widely studied for decades and remains the standard approach to many problems. Table 1 summarizes the most popular MARL benchmark environments that emphasize either collaborative toy problems (i.e, that usually support theoretical work) or realistic network control.

Many popular MARL suites (e.g., the Multi-Particle Environments, PySC2/SMAC, POGEMA) emphasize navigation, communication, or coordination in toy problems and are not primarily designed to evaluate distributed control on large networked systems. Conversely, realistic simulators that model networks are often oriented towards single-agent formulations or provide limited scalability/modularity for multi-agent experiments. To the best of our knowledge, there is currently

| Benchmark | Task type(s) | Max # agents | Modular | Partial Obs. |
|---|---|:---:|:---:|:---:|
| *Toy problems* | | | | |
| Multi-Particle Env. [13] | Navigation / Communication | 6 | ✗ | ✓ |
| Google Football [14] | Navigation / Control | 22 | ✗ | ✓ |
| LB-Foraging [15] | Navigation | $\sim$10 | ✓ | ✗ |
| SMACv2 [16] | Navigation / Control | $\sim$100 | ✗ | ✓ |
| POGEMA [17] | Navigation | $\sim$100 | ✓ | ✓ |
| MeltingPot [18] | Various | $\sim$10 | ✗ | ✓ |
| Multi-Agent Atari [19] | Various | 4 | ✗ | ✗ |
| SISL [20] | Navigation / Control | $\sim$8 | ✗ | ✓ |
| Overcooked [21] | Navigation / Control | 2 | ✗ | ✗ |
| **COGNAC** | **Network Control** | $\sim$**10 000** | ✓ | ✓ |
| *Realistic environments* | | | | |
| Grid2Op [22] | Network Control | $\sim$100 | ✗ | ✗ |
| WFCRL [23] | Network Control | $\sim$30 | ✓ | ✓ |
| SUMO-RL [24] | Network Control | $\sim$20 | ✗ | ✓ |

Table 1: Comparison of some popular multi-agent benchmarks. "Realistic" indicates the use of a physics-based or traffic-based simulator. Modular tasks indicate that the user has flexibility over the choice of problem size/structure and/or scenarios to play; "Partial Obs." denotes partial observability.

no benchmark environment suite dedicated to problems with network structure that emphasizes the development of distributed methods in multi-agent settings and can support theoretical work through modularity and simplicity of underlying dynamics. In addition, the work done on multi-agent networked systems and their evaluation is often carried out on custom problems or environments with no standard open-source implementation available [7, 5]. Therefore, **COGNAC** is designed as a benchmark for *network control* that (i) targets genuinely large-scale multi-agent problems (on the order of $10^2$ to $10^4$ agents), (ii) is modular so users can vary problem size and topology, and (iii) natively supports partial observability.

## 1.3 Motivations

The development of COGNAC for *COoperative Graph-based Networked Agent Challenges* is motivated by the lack of available RL benchmark environments dedicated to simple fully cooperative multi-agent problems with inherent graph structure. Specifically, the proposed package aims to provide a very flexible and modular platform to test control methods on various cooperative tasks, offering the possibility to run the environment on **any graph size** with **any structure**. This is especially helpful for the development of decentralized, federated, and hierarchical RL methods for problems that remain intractable with standard centralized methods.

The various environments proposed within the initial release of the package are designed to be as simple as possible in their formulation and dynamics, yet offer challenging tasks for model-free RL algorithms, even on smaller instances of problems. It can be used to evaluate the limitations of standard centralized RL methods on combinatorial control problems with exploding state and action space sizes. Overall, the lack of theoretical results in the literature is a primary concern motivating the development of **COGNAC**. Hence, our package provides a platform to address a key open challenge in this field: *Identify special structures that make problems theoretically solvable*. Therefore, we aim at closing the gap between theoretical work on network-based cooperative control problems and realistic simulation environments for real-life systems.

## Contribution of the Paper

- We introduce **COGNAC** (COoperative Graph-based Networked Agents Challenges)[1], a Python-based benchmark suite offering **flexible, graph-structured, cooperative multi-agent environments** for MARL research.

---

[1]https://github.com/yojul/cognac

- We provide **the first standardized open-source implementations** of several well-known theoretical graph-based MARL problems taken from the literature, adapted for empirical benchmarking with modern RL tooling.
- We offer a set of **baseline experiments with state-of-the-art MARL algorithms**[2], showcasing scalability challenges and opportunities for future method development.

## 2 COGNAC : Cooperative Graph-based Networked Agent Challenges

We consider fully cooperative multi-agent tasks with partial observability of the environment.

**MDPs and Decentralized Partially Observable MDPs**     The standard framework for reinforcement learning problems is to consider optimal control of a Markov Decision Process (MDP). A Markov Decision Process is defined by the tuple $\langle \mathcal{S}, \mathcal{A}, \mathcal{P}, \mathcal{R}, \gamma \rangle$, where $\mathcal{S}$ is the state space, $\mathcal{A}$ the action space, $\mathcal{P}$ defines the transition kernel over the state and action space, $\mathcal{R} : \mathcal{S} \times \mathcal{A} \rightarrow \mathbb{R}$ is the reward, and $\gamma \in [0, 1]$ is the discount factor of the problem. The standard objective in RL is to maximize the expected discounted return by solving the Bellman equation [25, 26]. In the context of MARL, this framework is extended to a Decentralized Partially Observable Markov Decision Process (DecPOMDP). A DecPOMDP is typically defined by a tuple $\langle \mathcal{N}, \mathcal{S}, \mathcal{A}, \mathcal{P}, \mathcal{Z}, \mathcal{O}, \mathcal{R}, \gamma \rangle$, where $\mathcal{N} = \{0, 1, ..., N\}$ is the set of $N$ agents. Each agent receives observations from its observation space $\mathcal{Z}$, which are generated according to an observation function or kernel $\mathcal{O}$ based on the environment state. The global objective is to find the optimal joint policy. In the general case, finding an optimal solution for a finite-horizon DecPOMDP is known to be NEXP-complete for $N \geq 2$ (i.e., two agents or more) [3].

### 2.1 Available Environments

The initial release of the package comes with 4 different problems: **Firefighting graph**, **Binary consensus**, **SysAdmin network**, and **Multi-commodity flow network**. These environments are inspired by classical pre-existing problems such as the SysAdmin network [27] or Firefighting Graph [6]. These problems have been widely studied and used as benchmark problems to test distributed multi-agent methods. However, to the best of our knowledge, there are no standard implementations available, which makes algorithm comparison more difficult in the long run. The chosen problems implemented in COGNAC are fully described in the following subsections. Some of them can be instantiated with any graph structure defined by the user, and we provide a collection of standard graph structures for benchmarking purposes with various sizes and properties: Directed Acyclic Graph, Tree, Undirected, Dense or Sparse graph, etc.

#### 2.1.1 Fire Fighting Graph

The Fire Fighting Graph problem was introduced by F. Oliehoek & C. Amato [6, 3] as a stylized version of a simple instance of a Dec-POMDP with a Dynamic Bayesian Network (DBN) structure. It can model various types of problems related to epidemic control [28], communication networks [29, 30], or traffic control [31]. In the original description of the problem, $N$ firemen are trying to extinguish fire on a row of $N + 1$ houses. This problem has a particular structure which can be modeled as a DBN. Specifically, it can be seen as a bipartite graph with $N$ agent nodes (the firemen) on the one hand and $N + 1$ environment nodes (the houses) on the other hand. This problem comes in two different versions in our package: the first one is the original as described, and we propose an extension of the problem to a grid problem with a $(N + 1) \times (M + 1)$ grid of houses controlled by $N \times M$ firemen. The following description applies to the original 1-dimensional environment, which is straightforward to extend to the 2-dimensional environment.

**State and Observation:** The state is described by the level of fire on each house as an integer with $s_h(t) \in [0, \theta - 1]$ and

$$S(t) = \{s_h(t)\}_{h=1}^{N+1}. \tag{1}$$

The state is not directly observed by the agents. Each agent observes the fire at its location (i.e., one of the houses it controls) with a specified probability that depends on the level of fire. This introduces

---

[2]https://github.com/yojul/cognac-benchmark-example

noise between the observed state and the actual state of the system. Implicitly, the environment also stores the location of the firemen to know which house each agent is observing.

**Action Space:** At each time step, agent $i$ can go to house $i$ or $i + 1$ to extinguish the fire with a certain probability. In the grid problem, an agent can go to one of the four houses around it.

**Objective and Reward Function:** An episode ends when the fire is fully extinguished in all houses, i.e., $S(t) = \{0\}_{h=0}^{N+1}$, or if it reaches the maximum horizon $t \geq T$. At each time step, the reward signal is given by the level of fire at the observed house at $t + 1$. Hence, the agents receive a reward based on their location, and the reward is linked to the houses:

$$r_i(t) = \begin{cases} -s_i(t) & \text{if agent } i \text{ is at house } i \\ -s_{i+1}(t) & \text{if agent } i \text{ is at house } i + 1 \end{cases} \tag{2}$$

The detailed description of the dynamics of the 1-D problem, including default transition probabilities, is given in [3]. The extension to the two-dimensional case is straightforward and fully described in the supplementary material.

### 2.1.2 Binary Consensus

This problem is directly inspired by the voter model introduced by Richard A. Holley and Thomas M. Liggett in 1975 as a simple model with a graph structure where each particle (or agent) can influence the vote of its neighbors [32]. While this type of modeling has been largely studied, especially in the field of statistical physics, there are very few adaptations of the voter model to Markov Decision Processes and, to the best of our knowledge, no available minimal formulation for a decentralized multi-agent setting with a fully cooperative objective. Therefore, we simply adapt this problem to make it a Dec-POMDP where each agent can observe its state and its neighborhood and act by either keeping its vote or changing it to the alternative value. In addition, the state of the agent is stochastically influenced by its neighbors' actions. Although very simple, this Dec-POMDP is non-trivial to solve.

Formally, $N$ agents vote simultaneously to keep or change their vote at each step and try to reach a consensus within a time horizon $T$. The graph structure of the problem describes the neighborhood of agents and quantifies how agents influence each other with their actions. The main difficulty when applying decentralized methods to this problem is that agents need to agree on a value by updating their local policies simultaneously. Thus, for decentralized methods, it is not obvious that algorithms can converge to a common value for consensus. Hence, we argue that this problem is interesting for comparison purposes and can be used as an interesting benchmark for convergence speed study.

**State and Observation:** Each agent maintains a state represented by a single boolean variable $s_i(t) \in \{0, 1\}$ corresponding to its vote at time $t$. The joint state at time $t$ is denoted

$$S(t) = \{s_i(t)\}_{i=1}^{N} \tag{3}$$

In the standard setting, each agent can observe its own vote as well as the votes of its neighbors as defined by the graph structure.

**Action Space:** At each step, an agent can choose to keep its current vote or change to the other value. Hence, we have

$$a_i(t) \in \{0, 1\}, \tag{4}$$

with 0 representing keeping its current vote, and 1 switching to the opposite vote.

**Objective:** The objective is to drive the system towards a consensus on any of the two values within the time horizon. Formally, let $m_t$ be the current majority vote at $t$ :

$$m_t = \arg \max_{v \in \{0,1\}} \sum_{i=1}^{N} \mathbb{1}[s_i(t) = v]. \tag{5}$$

A consensus is reached if all agents hold the same vote:

$$s_i(t) = m_t, \quad \forall i \in \{1, \ldots, N\}. \tag{6}$$

An episode ends when either a consensus is reached or $t > T$.

While this particular theoretical problem is not a direct representation of a real-world network system, its very basic formulation is very convenient to evaluate and compare centralized, semi-centralized, and decentralized methods with various network sizes and structures. It has a state space of size $2^N$ and an action space of the same size $2^N$, making the problem quickly intractable as the number of agents $N$ increases. This is particularly useful to assess the limitations of fully centralized methods. Since any consensus is a solution, once all agents have learnt to align on a particular value, the game is solved. An alternative setting for this problem is to make the agents aim at a particular consensus, for example, corresponding to the initial majority value $m_0$. This problem is way more difficult to solve in practice, but it is of interest.

### 2.1.3 Multi-agent SysAdmin

This particular problem was first introduced in 2002 by Guestrin et al. as a standard benchmark problem to evaluate planning methods leveraging network structure in factored Markov Decision Processes [27]. Initially defined as a single-agent problem, the multi-agent formulation was also introduced later by Guestrin et al. [5] to introduce coordinated RL methods in multi-agent settings. It has been studied and often referred to as a standard benchmark problem to this day [33, 34]. It has been treated in different settings and with some variations in the state space and definition of the dynamics. Here, we propose the first modern open-source standard implementation of the problem dedicated to MARL.

In this problem, a network of computers must achieve tasks, but each machine has a probability of becoming *faulty*, making the task longer to achieve, or even *dead*, making the task impossible to finish. When a computer encounters a fault, it has a probability of propagating the fault to its neighbors. At each timestep, the *SysAdmin* can choose to reboot some of the computers on the network; this will cause the loss of any progress on an ongoing task but will restore the computer to a working status *good* with high probability. The multi-agent formulation of the problem helps in assessing how the structure of the graph, and especially the relationship between neighbors, can be exploited to solve a collaborative task in the context of *Partially Observable Markov Decision Processes* (Pom-MDPs).

As with the consensus environment, the graph structure gives the topology of the network, quantifying how a fault of one agent can propagate to its neighbors.

**State Space:** Each agent maintains two features in its state: its working status as *good*, *faulty*, or *dead* and its load status, which can be *idle*, *loaded*, or *successful*. Formally, the joint state space is:

$$S(t) \in \{\text{good}, \text{faulty}, \text{dead}\}^N \times \{\text{idle}, \text{loaded}, \text{successful}\}^N \tag{7}$$

This gives the state space of cardinality $9^N$.

**Action Space:** At each time step, each agent can choose to do nothing or to reboot for the next time step. Rebooting will cause the computer to lose progress on its current task, but will reset the computer to the working status *good* with high probability.

**Objective:** The objective is to maximize the number of solved tasks. Theoretically, the problem can be studied both as an infinite horizon problem and a finite horizon problem with horizon $T$.

### 2.1.4 Multi-commodity Flow

Network flow problems are a class of combinatorial optimization problems defined on a graph, where the goal is to optimize the distribution of flow along edges under specific constraints. Many variants of these problems exist, as extensively covered in [35]. Here, we implement a specific version of the multi-commodity flow problem in a multi-agent setting with partial observability. In this problem, the objective is to minimize the total cost of flow circulation on the network, given that edges have costs and capacities. Multi-commodity refers to multiple classes of flow available in the problem. Even with only two commodities, finding an optimal integer flow that satisfies the constraints is NP-complete.

Our implementation can handle both the flow problem involving sources and sink nodes and the particular case of the circulation problem where the network is initialized with only circulation nodes and initial flows on those nodes. During an episode, the flows must satisfy several constraints:

- **Edge capacity**: The total flow on an edge between nodes $i$ and $j$ cannot exceed its capacity $\rho_{ij}$. Let $\rho_{max} = \max_{(i,j) \in N^2} \rho_{ij}$ be the maximum capacity on the network.

- **Flow conservation**: For any circulation node, the total incoming flow must equal the total outgoing flow.
- **Flow conservation at source and sink nodes**: A commodity must fully exit its source and fully enter its sink.

Here, we consider the multi-agent problem where each node of the network is an agent that needs to handle how it dispatches the incoming flows to its outgoing edges. Thus, each agent observes the flows arriving at its incoming edges and decides how to dispatch them to its outgoing edges. The environment only allows flows with integer values, making the action space discrete by definition.

**State Space:** We consider a directed graph where each node has at least one incoming and one outgoing edge. Let $k$ be the number of commodities (flow classes), $N$ the number of controllable circulation nodes (i.e., agents), and $E$ the number of edges in the network. At each timestep, the global state consists of the flow values on each edge. Each agent observes the flow values on its incoming edges only.

**Action Space:** The individual action space of each agent is defined as a vector (or matrix) describing the dispatch of flows to outgoing nodes. Hence, for each agent $i$, it has $k \times n_{out}^i$ elements. Flows are integers; however, in practice, an agent's policy predicts a distribution of flow along edges that is mapped to an integer dispatch on outgoing edges.

**Objective:** Here, agents need to minimize the total cost of flows circulating on the network. Each class of flow (commodity) and edge has an associated cost. The objective is then to minimize the total circulation cost of flows on the network during an episode with horizon $T$.

### 2.2 Multi-Agent Reinforcement Learning with COGNAC

We designed the `COGNAC` package to be as easy to use as possible. Each problem is implemented as a self-contained `PettingZoo` environment, making it directly compatible with most existing implementations of MARL algorithms. The environments are easy to run, with clear and understandable dynamics. A key feature of the package is its modularity. Users can easily customize environment parameters to adjust the difficulty or explore different settings. All environments support variable sizes, and **Binary Consensus** and **SysAdmin** are graph-agnostic, allowing them to be initialized with any weighted adjacency matrix. Table 2 provides an overview of the joint state and action space sizes, as well as the modular features of each environment.

Table 2: Features and size comparison of multi-agent environments available in COGNAC

| Environment | Modular Size | Graph Agnostic | Joint State Space | Joint Act. Space |
|---|---|---|---|---|
| Firefighting Graph (1D) | ✓ | ✗ | $\theta^N$ | $2^N$ |
| Firefighting Graph (2D) | ✓ | ✗ | $\theta^{N \times M}$ | $4^N$ |
| Binary Consensus | ✓ | ✓ | $2^N$ | $2^N$ |
| SysAdmin | ✓ | ✓ | $9^N$ | $2^N$ |
| Multi-commodity Flow | ✓ | ✗ | $\rho_{max}^{k \times E}$ | $\rho_{max}^{k \times E}$ |

The various benchmark environments provided offer increasing combinatorial complexity in terms of joint action and state space sizes. The graph structure also differs from one problem to another, especially in the way agents or the environment influence the dynamics. Thus, the implemented environments offer a variety of network-based dynamics, which can be very convenient for comparing the efficiency of multi-agent methods in various contexts. For example, in the **Firefighting Graph** problems, the network structure determines the noisy observation and action space available to the agent, and therefore creates indirect dependencies between neighboring agents. In addition, the network structure of the environment is decorrelated from the relationship between agents. In the **Binary Consensus** problem, the dependency is more direct, and there is a single graph structure (i.e., the neighborhood relationship between agents), since the action of an agent will directly and stochastically influence the state transitions. Overall, the environments offer different challenges: **Binary Consensus** and **Firefighting Graph** are meant to be solved as fast as possible to maximize reward, while **SysAdmin** and **Multi-commodity Flow** can be interpreted as infinite-horizon control tasks in their default settings.

In addition, we provide some basic rendering functionalities in order to visualize the environments. This is particularly helpful to qualitatively compare multi-agent policies or assess whether learned policies act as expected. This can be used to identify unexpected behavior directly by observing some trajectories.

## 3   A Benchmark Example

We propose a simple benchmark example of some standard decentralized learning algorithms on COGNAC environments. We test two independent learning algorithms: Independent Deep Q-Learning (IDQL) and Independent Proximal Policy Optimization (IPPO) [36]. Independent learning refers to algorithms where each agent learns its own policy independently, treating other agents as part of the environment without explicit coordination [37]. In addition, we implement their respective adaptation in the paradigm of Centralized Training Decentralized Execution (CTDE): Q-Mix [38] and Multi-Agent Proximal Policy Optimization (MAPPO) [39]. For CTDE algorithms, policies are not shared: each agent maintains its own policy. The results obtained are reported in Figure 1 and compared with heuristic baselines. Our implementations of these algorithms are directly adapted from the *CleanRL* implementations [40].

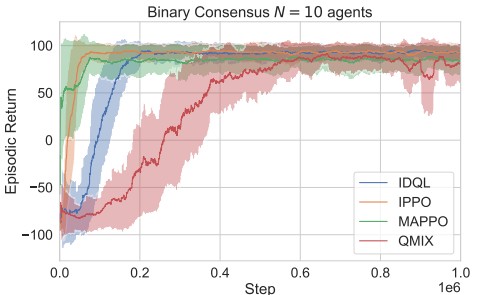

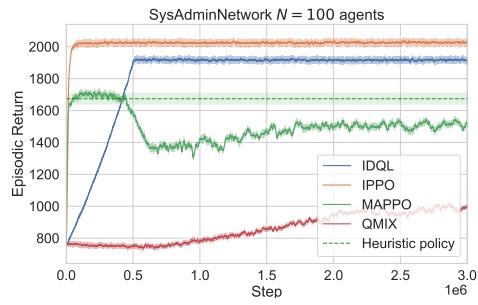

(a) No heuristic policy as the optimal policy in this setup corresponds to a policy where every agent always aims at voting for one of the two values.

(b) Heuristic policy corresponds to taking reboot action on a machine whenever the process is dead.

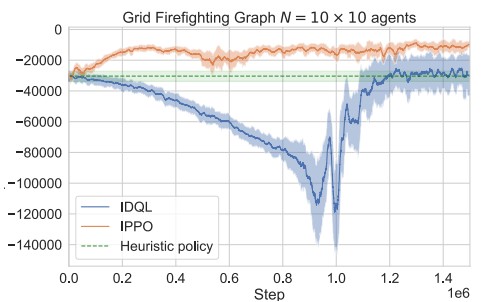

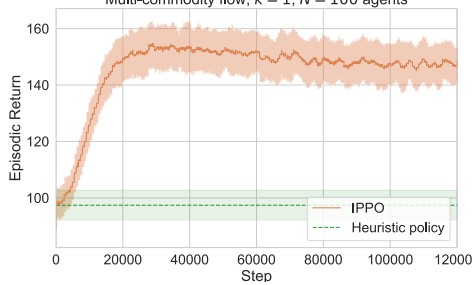

(c) Heuristic policy corresponds to choosing one of the four observable houses randomly.

(d) Heuristic policy corresponds to a centralized linear program (LP) that maximizes the total flow utility across the network, based on edge costs.

Figure 1: Training comparison on COGNAC's environment with various sizes and heuristic baselines. The shaded area corresponds respectively to the standard deviation of the return obtained on a large number of episodes for heuristic policies (green) and to the standard deviation of the running average of the return obtained for each algorithm. For the commodity circulation problem (1d), individual action spaces are considered as continuous, hence IDQL is not suitable for this environment.

Despite the non-stationarity inherent to the Dec-POMDP formulation of these problems, the IPPO algorithm consistently learns effective policies across all tested environments. In the **Binary Consensus** environment, IDQL reaches comparable final performance but requires substantially more training steps, while it performs worse in the other discrete tasks. In this same environment, centralized training approaches (MAPPO and QMIX) eventually reach consensus, yet their convergence is slower and less stable. Moreover, these CTDE-based methods fail to generalize effectively to larger or more interdependent systems, indicating limited scalability.

In the **SysAdmin** environment with 100 agents, MAPPO and QMIX perform significantly below independent methods, particularly IPPO, which rapidly converges to a near-optimal policy and clearly outperforms the heuristic baseline. While extended training might allow CTDE methods to approach the heuristic policy's performance, the observed results emphasize their difficulty in scaling to large agent populations with strong inter-agent dependencies.

For the **Grid Firefighting** task, IPPO achieves stable improvement and surpasses the random heuristic baseline, while IDQL struggles with the combinatorial complexity. In the **Multi-Commodity Flow** problem, where the action space is defined as continuous, we only apply an adapted version of IPPO, and it successfully converges towards a high-performing policy that outperforms the centralized linear program heuristic.

We use standard hyperparameter sets that can be found in the literature and perform minimal tuning, with similar configurations used across all environments. The empirical evidence suggests that IPPO offers superior out-of-the-box performance for decentralized control problems. These findings align with prior observations by [36, 23], reinforcing the idea that independent actor-critic methods can provide a naive yet effective baseline for large-scale cooperative MARL tasks.

Decentralized algorithms appear to be a natural choice for this type of problem, but they have to be compared against fully centralized methods, which are much more straightforward to apply and have theoretical convergence guarantees. Figure 2 shows a comparison of a fully centralized PPO algorithm against fully decentralized IPPO on two instances of the **Binary Consensus** environment with size $N = 10$ and $N = 100$ agents. In the centralized setting, a global reward signal is computed as the sum of local rewards.

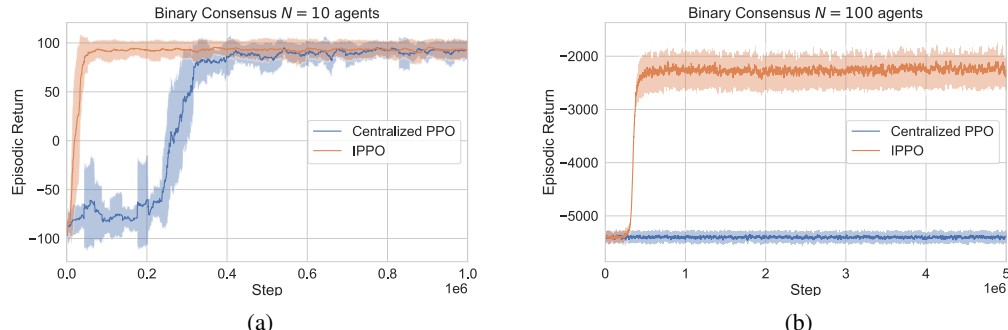

Figure 2: Comparison of episodic return of learned policies for fully centralized PPO algorithm against IPPO (fully decentralized) on two instances of size $N = 10$ (2a) and $N = 100$ (2b) on the Binary Consensus environment. The episodic return is computed as a moving average, and the shaded area corresponds to the standard deviation.

While centralized PPO is able to learn a good policy on a small instance of the problem with $N = 10$ agents, it fails when scaling up to $N = 100$ agents. This can be explained by the combinatorial aspect of the problem, as the action space is represented with a binary vector of size $N$. Even with complete observation of the problem, the centralized model-free approach fails to learn a working policy within reasonable learning times. Meanwhile, the IPPO algorithm scales successfully on larger instances of the problem with $N = 100$ and learns faster on the smaller instance. Furthermore, we observe a more stable learning curve on the small instance with $N = 10$ for IPPO.

**Additional experiments**   We propose a set of additional results in Appendix C with experiments on larger systems using independent learning methods, as well as a network density influence study and an enhanced PPO-based methods comparison on the SysAdmin problem using shared actor-policy and GNN-based aggregator (InforMARL algorithm [41]).

# 4   Limitations

As highlighted in the introduction section, COGNAC aims at bridging the gap between theoretical results on distributed control in MARL settings and applied research on network systems. Indeed, the implementation of environments is made as simple as possible without compromising the modularity and flexibility needed for research purposes. All environments are scalable to any size, and **Binary Consensus** and **SysAdmin** can be instantiated with any network structure, which makes them highly suitable to study and compare methods against particular network structures and properties. **Multi-commodity flow** remains the most difficult to solve as it has a very large combinatorial action space. While these benchmark environments provide interesting challenges to test and compare algorithms, it is generally difficult to compute theoretically optimal policies to compare with. This can be a drawback from the theoretical perspective, as a heuristic-based policy might not be a sufficient baseline. Furthermore, as the dynamics of environments are directly implemented in Python without relying on any third-party backend (e.g., directly in C or C++), simulating very large systems can become computationally expensive, especially without parallelization.

Regarding the benchmark example, the results highlight the efficiency of the IPPO algorithm in each environment tested and suggest that IPPO might be a very versatile choice as a fully decentralized independent learning method. Despite our best efforts in tuning CTDE algorithms and especially MAPPO, we observe that standard versions of these methods struggle when scaling up problems with strong inter-agent dependencies. This suggests that the CTDE paradigm might not be suitable to tackle the complexity and non-stationarity of large-scale problems, and that having a centralized component (here, the value function or critic) still faces the curse of dimensionality.

However, we recognize that methods with additional mechanisms that leverage communication and structure between agents [42, 43] may be able to perform well. Especially, implementing GNN-based methods such as the InfoMARL algorithm [41] is in line with future work and improvement of our benchmark. An extended benchmark on the SysAdmin Network problem comparing InforMARL with IPPO and standard MAPPO implementations with both independent and policy-sharing is available in Appendix C. This will be extended to other environments in future work. Recent developments in RL with GNNs have shown promising results on a wide range of applications [44]. There are many possible approaches to network and graph-based problems. Hence, we argue that COGNAC can stand as a solid benchmark platform for testing and comparing state-of-the-art methods, ranging from centralized standard and advanced methods, such as GNNs, to fully distributed independent learning methods.

# 5   Conclusion

We introduced COGNAC, the first collection of benchmark environments for MARL dedicated to cooperative tasks with network structures. The collection comprises four distinct types of problems directly taken from or inspired by theoretical benchmark problems from the literature. The environments are implemented in a fully self-contained manner, with minimal structural complexity and no reliance on external backend simulators. This makes the usage highly straightforward and flexible for research purposes. In addition, all environments are built to be as modular as possible, enabling the study of scalability with respect to problem size and graph structure. We provide a benchmark example with four of the most popular MARL methods to demonstrate how it can be used to compare algorithms in a simple and flexible way. In spite of their simplicity, we show empirically that independent learning algorithms such as IDQL and IPPO can be very efficient, overcoming the inherent non-stationarity induced by the DecPOMDP structure of the problem. COGNAC is released as an open-source package, along with the benchmark algorithms and trained models. We argue that COGNAC can be a valuable intermediate testbed platform for distributed MARL methods at the frontier between theoretical work on DecPOMDPs and applications on network systems. We hope that COGNAC will help advance research in distributed reinforcement learning by emphasizing the role of network topology in learning dynamics.

## Acknowledgments and Disclosure of Funding

This work is carried out in the framework of AI-NRGY (Grant ref.: ANR-22-PETA-0004) and REDEEM (Grant ref.: ANR-23-PEIA-005) projects funded by France 2030.

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

# A  Practical Information on COGNAC Package

The code for the environment, as well as some utility functions used to manipulate the environment and generate adjacency matrices and networks, is available at `https://github.com/yojul/cognac`. Cognac is released as open-source under the Apache 2.0 license, available at `https://github.com/yojul/cognac/blob/main/LICENSE.txt`. The complete readable documentation of the package can be found at `https://cognac-marl.readthedocs.io/en/latest/`. The environment code is also released as a PyPI package at `https://pypi.org/project/cognac/` and can be installed as any standard package using `pip install cognac`.

# B  Details on Environment Dynamics

We provide detailed descriptions of the environment's dynamics. In addition to the fully commented open-source code, we also provide here the step-function algorithm for each environment in natural language. This should help users understand the dynamics of each environment and make it easier to modify.

**Note**: The algorithms described here may be improved through ongoing research on Decentralized MARL, particularly by enhancing the dynamics and implementing additional mechanisms within the open-source project COGNAC.

## B.1  Firefighting Graph

Table 3 summarizes the different cases for fire dynamics for increasing and decreasing the fire level. Table 4 gives the individual observation probabilities. Figure 3 shows an illustration of both the 1-dimensional and the 2-dimensional problems, where agents can respectively observe 2 and 4 houses. The extension from the 1-dimensional problem with a row of houses and agents to the 2-dimensional problem as a grid induces one main difference regarding the size of individual observation and action spaces.

**State and Observation:** The state is described by the level of fire in each cell of a 2D grid as an integer with $s_{i,j}(t) \in [0, \theta]$, and

$$S(t) = \{s_{i,j}(t)\}_{i=0}^{N}{}_{j=0}^{M}. \tag{8}$$

**Action Space:** Each agent $i$ can go to one of the four houses around him as represented in Figure 3. It can reach houses $h_{i,j}, h_{i,j+1}, h_{i+1,j}, h_{i+1,j+1}$ and actions are encoded with an integer from 0 to 3 accordingly.

**Objective and Reward function:** The global objective remains the same, and the reward is still computed based on the level of fire at the last visited house.

| Category | Case | Visited? | Burning Neighbors? | Burning? | Fire Level | Default Probability |
|----------|------|----------|--------------------|----------|------------|---------------------|
| **Increasing** | Case 1 | ✗ | ✓ | ✓ | +1 | 0.7 |
| | Case 2 | ✗ | ✓ | ✗ | +1 | 0.3 |
| | Case 3 | ✗ | ✗ | ✓ | +1 | 0.4 |
| **Decreasing** | Case 4 | ✓($\geq 2$) | – | – | $\to 0$ | 1.0 |
| | Case 5 | ✓($= 1$) | ✓ | ✓ | -1 | 0.6 |
| | Case 6 | ✓($= 1$) | ✗ | ✓ | -1 | 1.0 |

Table 3: Fire level update rules in the Firefighting simulation environment. Default probabilities are taken from [3]. These can be easily modified to adjust the difficulty.

| Condition | | Default Observation Probability |
|---|---|---|
| The house is **not burning** | `fire_level = 0` | 0.2 |
| The house is **burning** | `fire_level = 1` | 0.5 |
| The house is **burning** | `fire_level > 1` | 0.8 |

Table 4: Observation probabilities of flames by agents. Default probabilities are taken from [3].

---

**Algorithm 1** Step Function of the `FirefightingGraphEnvironment` (1D Row and 2D Grid)

---

**Require:** Actions from all agents: `actions`: $\{0,1\}^N$ (1D) or $\{0,3\}^{N \times M}$ ▷ For agent $i$, action $a_i$ is encoded as an integer.
 1: Update house visits based on `actions`.
 2: **for** each case in Table 3 **do**
 3:     Filter houses matching the case conditions.
 4:     Apply the corresponding fire level change with the specified probability.
 5: **end for**
 6: Increment timestep: `timestep` ← `timestep` + 1
 7: Determine termination condition:

$$\text{done} \leftarrow (\text{state} = \mathbf{0})$$

 8: Determine truncation:

$$\text{truncated} \leftarrow (\text{timestep} > \text{max\_steps})$$

 9: Compute rewards:
10: `rewards` ← `reward(actions, state, done, truncated)`
11: Generate new observations based on the current `state`, using the rules from Table 4.
12: **return** `observations, rewards, done, truncated, info`

---

## B.2 Binary Consensus

The algorithm for the state update of the **BinaryConsensus** environment is fully described in Algorithm 2. In addition, Figure 4 gives an illustration of the dynamics. Finally, we provide some details and insights on the environment dynamics as well as some suggestions on extended usage.

---

**Algorithm 2** Step Function of the BinaryConsensusNetworkEnvironment

---

**Require:** Actions from all agents $\mathtt{actions} : \{0, 1\}^N$
1: Sample an influence activation matrix $I \in \{0, 1\}^{N \times N}$ such that:

$$I_{i,j} \sim \text{Bernoulli}(P_{i,j}) \quad \text{for all } i, j$$

where $P = |\mathtt{adjacency\_matrix}|$ is the matrix of influence probabilities.
2: **for** each agent $i \in \{1, \dots, n\}$ **do**
3:     Let $a_i \leftarrow \mathtt{actions}[i]$
4:     Initialize influence counter: $\Delta \leftarrow (-1)$ if $a_i = 0$, else $\Delta \leftarrow 1$
5:     **for** each agent $j \in \{1, \dots, n\}$ such that $I_{i,j} = 1$ **do**
6:         **if** $\mathtt{actions}[j] = 1$ **then**
7:             $\Delta \leftarrow \Delta + \text{sign}(\mathtt{adjacency\_matrix}[i, j])$
8:         **else**
9:             $\Delta \leftarrow \Delta - \text{sign}(\mathtt{adjacency\_matrix}[i, j])$
10:         **end if**
11:     **end for**
12:     Update action $a_i$ based on influence:

$$a_i \leftarrow \begin{cases} \text{random choice in } \{0, 1\} & \text{if } \Delta = 0 \\ 1 & \text{if } \Delta > 0 \\ 0 & \text{if } \Delta < 0 \end{cases}$$

13:     Update internal state:
$$\mathtt{state}[i] \leftarrow |\mathtt{state}[i] - a_i|$$

14: **end for**
15: Increment timestep: $\mathtt{timestep} \leftarrow \mathtt{timestep} + 1$
16: Determine if consensus is reached:

$$\mathtt{done} \leftarrow \left( \mathtt{state} = \vec{0} \text{ or } \mathtt{state} = \vec{1} \right)$$

17: Determine if truncated:

$$\mathtt{truncated} \leftarrow (\mathtt{timestep} > \mathtt{max\_steps})$$

18: Compute rewards via reward model:
19: $\mathtt{rewards} \leftarrow \mathtt{reward}(\mathtt{actions}, \mathtt{state}, \mathtt{done}, \mathtt{truncated})$
20: Generate new observations from the current state.
21: **return** Observations, Rewards, Terminations, Truncations, Info

---

Algorithm 2 shows that the state update of each agent is equally influenced by its own action as well as the actions of each of its neighbors. Therefore, it suggests that a denser network with more edges may make consensus more difficult to reach as the number of edges in the network increases. In addition, note that the definition of the action as *Keep* or *Change* the vote implies that the state itself is not directly considered, adding difficulty when there are strong interactions between agents.

## B.3 SysAdmin

Algorithm 3 describes the dynamics for the step method of the **SysAdminNetwork** environment. This environment is arguably the easiest to solve in COGNAC, and it can be observed empirically. This environment is very convenient to test the scalability of decentralized methods, and we were able to scale up independent learning algorithms up to randomly generated graphs with 10,000 nodes (i.e, agents) and up to $\sim 10^6$ edges. Results are reported in Figure 8.

| Parameter | Default Value | Description |
|---|---|---|
| `adjacency_matrix` | — | Graph structure defining agent connectivity |
| `base_arrival_rate` | 0.5 | Base rate for task arrivals |
| `base_fail_rate` | 0.1 | Base failure probability for agents |
| `dead_rate_multiplier` | 0.2 | Multiplier for death rate when failing |
| `base_success_rate` | 0.3 | Probability of success for healthy agents |
| `faulty_success_rate` | 0.1 | Probability of success for faulty agents |

Table 5: Default parameters for the SysAdmin environment.

---

**Algorithm 3** Step Function of the SysAdminNetworkEnvironment

---

**Require:** Actions from all agents $\texttt{actions} : \{0,1\}^n$

1: **Reboot machines**: Set to working and idle where $\texttt{actions}_i = 1$

$$\texttt{state}[i,0] \leftarrow 0, \quad \texttt{state}[i,1] \leftarrow 0 \quad \text{if } \texttt{actions}_i = 1$$

2: **Solve tasks for working machines** with loaded jobs

$$\texttt{state}[i,1] \mathrel{+}= \texttt{Binomial}(1, \texttt{base\_success\_rate}) \quad \text{if } \texttt{state}[i,0] = 0 \wedge \texttt{state}[i,1] = 1$$

3: **Solve tasks for faulty machines** with loaded jobs

$$\texttt{state}[i,1] \mathrel{+}= \texttt{Binomial}(1, \texttt{faulty\_success\_rate}) \quad \text{if } \texttt{state}[i,0] = 1 \wedge \texttt{state}[i,1] = 1$$

4: **Reset completed tasks**

$$\texttt{state}[i,1] \leftarrow 0 \quad \text{if task done at } i$$

5: **Sample new jobs for available machines**

$$\texttt{state}[i,1] \mathrel{+}= \texttt{Binomial}(1, \texttt{base\_arrival\_rate}) \quad \text{if } \texttt{state}[i,0] = 0 \wedge \texttt{state}[i,1] = 0$$

6: **Induce failures on working machines via network influence**

$$\texttt{state}[i,0] \leftarrow 1 \quad \text{with prob. } \sum_j \texttt{adj\_matrix\_prob}[i,j]$$

7: **Induce dead states on faulty machines via network influence**

$$\texttt{state}[i,0] \leftarrow 2 \quad \text{with prob. } \texttt{dead\_rate\_multiplier} \cdot \sum_j \texttt{adj\_matrix\_prob}[i,j]$$

8: Increment timestep: $\texttt{timestep} \leftarrow \texttt{timestep} + 1$
9: Determine if consensus is reached:

$$\texttt{done} \leftarrow ((\texttt{state} = 0, \forall i \in \{1, \ldots, N\}) \text{ or } (\texttt{state} = 1, \forall i \in \{1, \ldots, N\}))$$

10: Determine if truncated:

$$\texttt{truncated} \leftarrow (\texttt{timestep} > \texttt{max\_steps})$$

11: Compute rewards via reward model:
12: $\texttt{rewards} \leftarrow \texttt{reward(actions, self, done, truncated)}$
13: Generate new observations based on current state.
14: **return** Observations, Rewards, Terminations, Truncations, Info

---

## B.4 Multi-commodity Flow Network

This environment is arguably the most difficult to solve. The original problem of multi-commodity flow with integer flows is known to be NP-hard. Here, the multi-agent version of the problem aims at solving the problem in a decentralized way. Figure 6 illustrates the environment from the perspective of an agent within the network. Algorithm 4 describes the dynamics of the environment in the initial release of the package. For convenience purposes, we choose to formulate the action as a distribution of the dispatch for each commodity on each of its output edges. For simplicity, the environment can be modified to process continuous flows but we aim at solving the integer flow problem. The implementation and formulation of this environment might evolve in future work.

---

**Algorithm 4** Step Function of the MultiCommodityFlowEnvironment

---

**Require:** Actions from all agents `actions` $: \mathbb{R}^{n \times m_i}$ with $m_i$ the number of outgoing edges from agent $i$
 1: Increment timestep: `timestep ← timestep + 1`
 2: **for** each agent $i \in$ `actions` **do**
 3:     **if** `len(actions[i])` $= 0$ **then**
 4:         **continue**
 5:     **end if**
 6:     Normalize action into distribution: `distribution ← actions[i]/` $\sum$ `actions[i]`
 7:     Compute dispatchable stock: `stock ← min(max_capacity, commodities[i])`
 8:     Update node stock: `commodities[i] ← commodities[i] − stock`
 9:     Split stock across outgoing edges using distribution:

$$\texttt{dispatch} \leftarrow \texttt{split\_integer\_by\_distribution(stock, distribution)}$$

10:     Set edge flows: $\texttt{flow}_{i \to j} \leftarrow \texttt{dispatch}_j$
11: **end for**
12: **for** each agent $i$ **do**
13:     Update commodities with incoming flows:

$$\texttt{commodities}[i] \leftarrow \texttt{commodities}[i] + \sum_{j \to i} \texttt{flow}_{j \to i}$$

14: **end for**
15: Generate new observations from updated state
16: Compute rewards:
17: `rewards ← reward(actions, env, False, False)`
18: Initialize terminations and truncations:
19: `terminations ← False for all agents`
20: `truncations ← False for all agents`
21: **if** `timestep` $\geq$ `max_steps` **then**
22:     `truncations ← True for all agents`
23:     `terminations ← True for all agents`
24:     `rewards ← reward(actions, env, True, True)`
25:     `agents ←` $[\,]$
26: **end if**
27: Verify total circulation is conserved:

$$\sum_i \texttt{commodities}[i] = \texttt{initial\_total\_circulation}$$

28: **return** Observations, Rewards, Terminations, Truncations, Info

---

## B.5 Illustrations

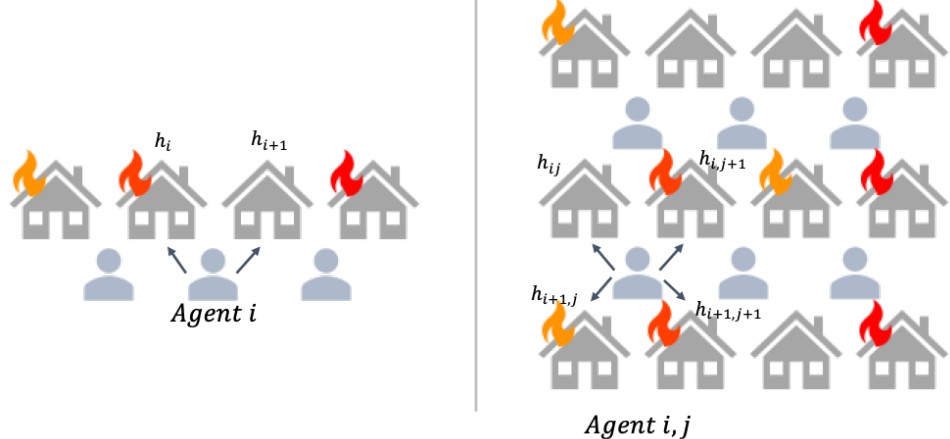

Figure 3: Illustration of the *FirefightingGraph* problem in 1-Dimension (left) and its natural extension to the grid problem in 2D (right).

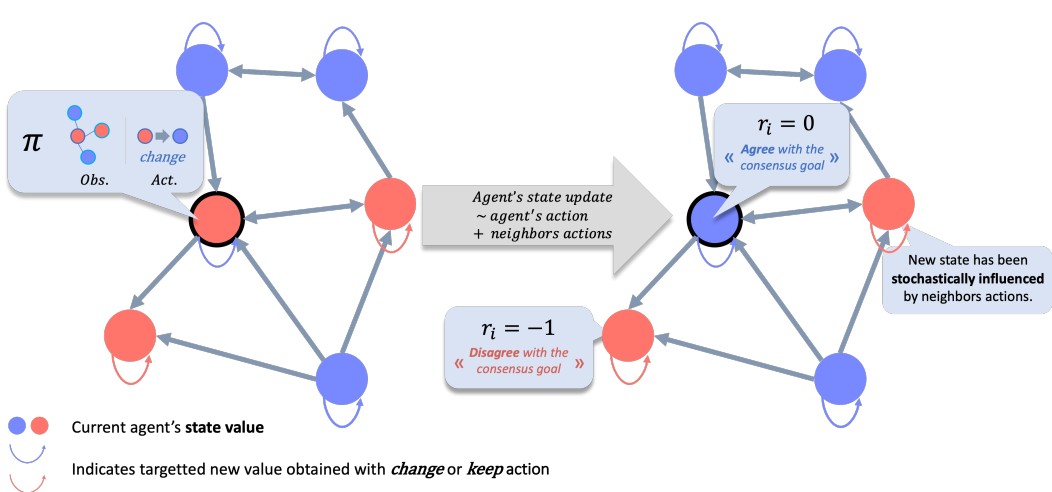

Figure 4: Illustration of the *Binary Consensus* problem.

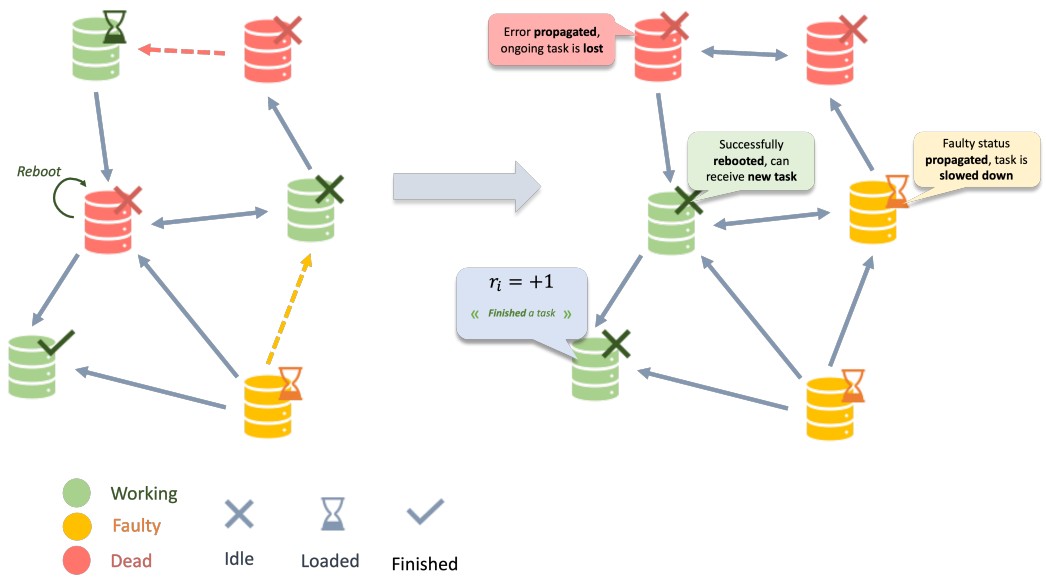

Figure 5: Illustration of the *SysAdmin Network* problem.

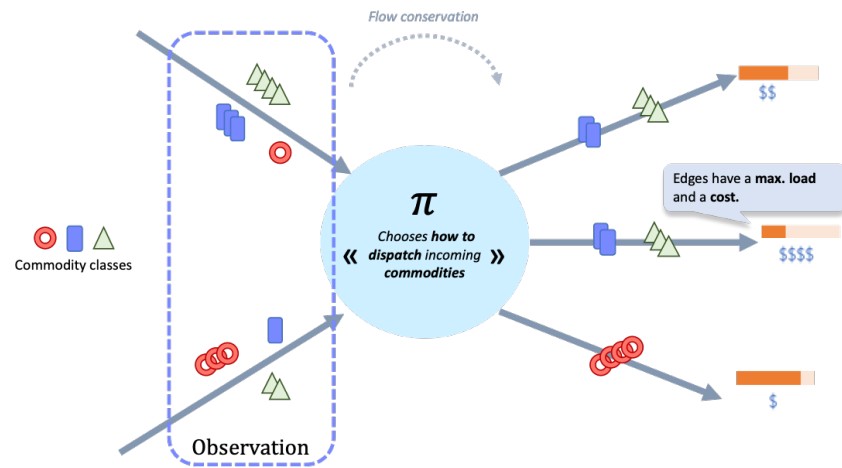

Figure 6: Illustration of the *Multi-commodity flow* problem.

# C Additional results

## C.1 Large systems

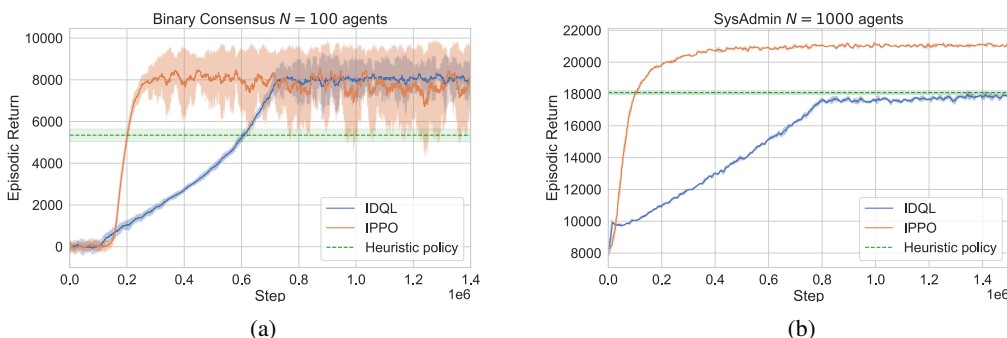

(a)        (b)

Figure 7: Comparison of independent learning algorithms on larger systems.

On Figure 7, we consider instances of Binary Consensus and SysAdmin ten times larger than the results presented in the benchmark example of the main text of the paper (Figure 1. Here, we do not consider CTDE algorithms (i.e MAPPO and Q-MIX) as these algorithms are much slower to run. Therefore, we were unable to get convergence in a reasonable time. This is mainly because of the centralized critic/value function. While we observe that these algorithms are still able to converge towards a solution in binary consensus, the convergence is very slow due to the dimensionality of the centralized view in these algorithms. These results on a larger system allow a broader comparison of independent learning algorithms and showcase the superiority of IPPO on both problems in terms of convergence speed and final decentralized joint-policy on the SysAdmin problem.

On Figure 8, we try to learn a decentralized joint-policy with an independent learning algorithm on a network with $10,000$ agents and $10^6$ edges (i.e, strong and multiple dependencies).

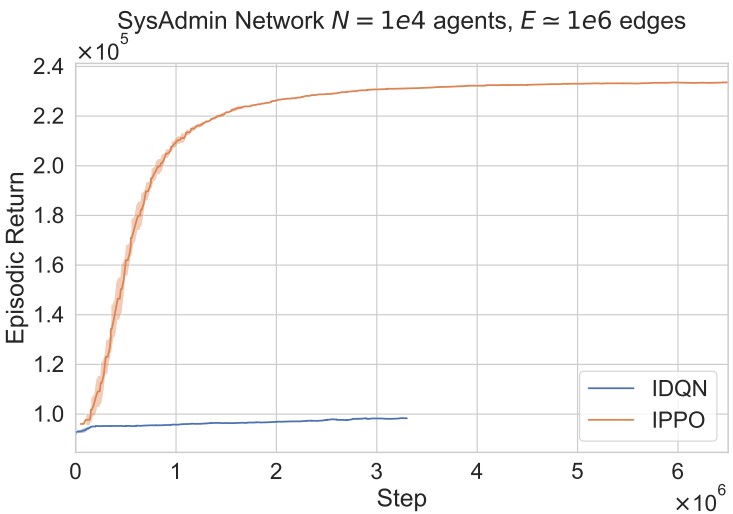

Figure 8: Example of training for independent learning algorithms on a large-scale instance of the SysAdminNetwork problem. Here, the network has 10000 agents and $\sim 10^6$ edges. The graph is generated randomly, with each potential edge between two nodes included independently with probability 0.1. Edge weights are assigned independently from a uniform distribution. IDQL was stopped early due to very slow convergence as compared to IPPO and heavy computation load.

On the one hand, we observe that the IPPO algorithm still manages to learn an efficient policy even for a larger system without any hyperparameter tuning on this particular problem. On the other hand, as observed in other experiments, the IDQL algorithm struggles with slow learning problems even

with some standard hyperparameter tuning. We expect that more advanced Value-based methods such as QMIX [38] or Value-Decomposition-Network [45], could reach or even be better than the standard IPPO algorithm.

## C.2  Network density

As one of the main objectives of our package is to study the influence of the network's structure and properties on the learning process, Figure 9 shows the evolution of the average final reward as a function of the graph density. To do this, we generate Erdős–Rényi graphs with increasing probability (i.e., the probability that there is an edge with non-zero influence probability between two nodes). We start from a graph with no edge and increase connectivity up to a fully connected graph. The influence probabilities on the edges are drawn from a uniform distribution between 0 and 1.

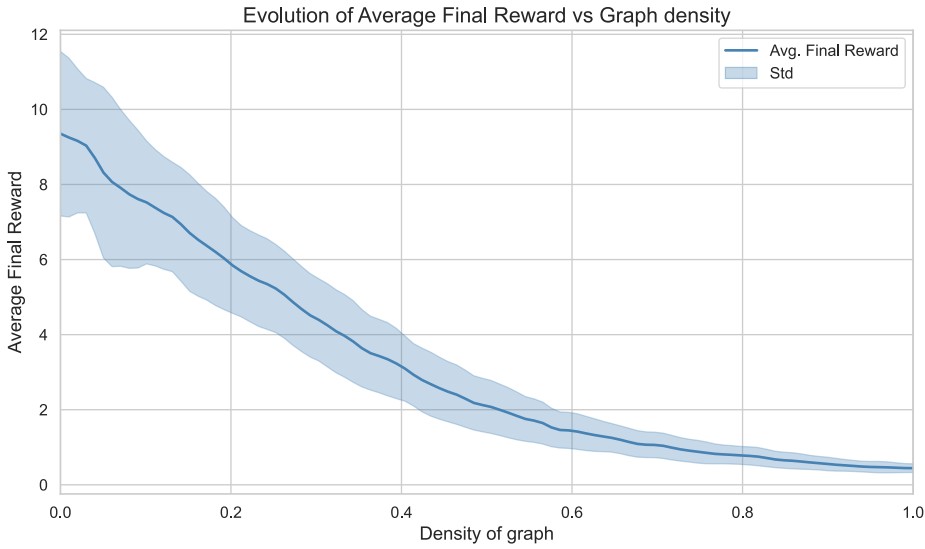

Figure 9: Average final reward achieved by standard IDQL training as a function of the density of randomly generated graph structures with $N = 10$ agents. For a given density, the average final reward is computed over 100 Erdős–Rényi graphs. For each structure, we evaluate the performance of the trained policy over 100 episodes. The training process remains the same for all tested graphs.

The results suggest that denser networks make the environment more difficult to solve with standard training procedures. The binary consensus environment is well-suited for this type of experiment, as it features binary state and action spaces while remaining quite challenging due to the strong interactions between agents. Indeed, it is worth noting that when interactions are made deterministic (i.e., edge weights set to 1), the optimal policy is not trivial to find—even for a human player—and some initial configurations cannot be solved. Moreover, this also depends on the time horizon of the environment, as solving it may require a complex sequence of coordinated actions.

## C.3  PPO-based methods and GNN extension

The **SysAdmin** problem has homogeneous agents, i.e, they all have the same observation and action space. The interdependency between agents is implicit and not explicitly described in the agent's observation space. **MAPPO** algorithms usually have the best results using a policy sharing mechanism across agents in homogeneous settings [39]. In addition to being clearly faster, it also allows the use of experience buffers from all agents to train the actor policy. In Figure 10, we compare Independent PPO, where each agent maintains its own actor-critic policy, to 2 standard **MAPPO** implementations and the **InfoMARL** algorithm. **InforMARL** uses Graph Neural Networks to aggregate information from each agent's neighborhood. This allows an efficient use of the network structure of the problem [41]. However, this also supposes additional information for each agent that uses its own observation of the system and the output of the GNN aggregator. The aggregator takes into account neighborhood information and allows for an efficient use of a shared actor policy across agents.

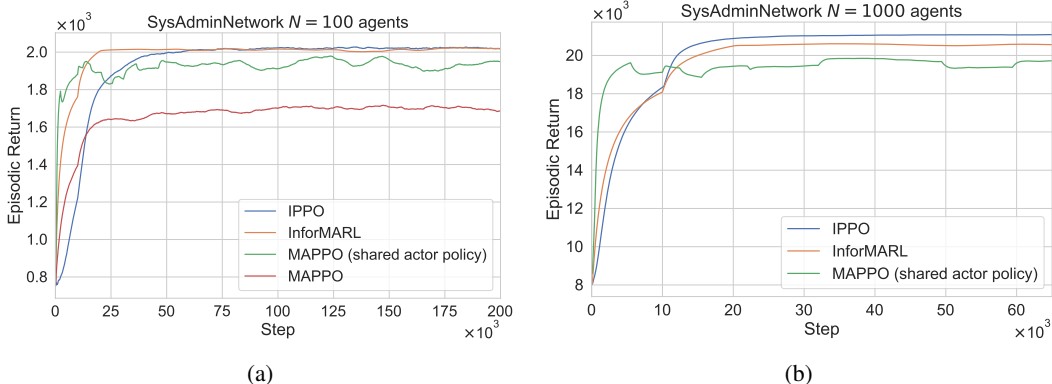

(a)                                                                    (b)

Figure 10: Comparison of PPO-based methods on the SysAdmin Network problem. We compare Independent PPO to three different versions of MAPPO: 2 standard implementations with shared and independent actor-policy, and InforMARL algorithm that uses a GNN-based aggregator for both actor and critic network. MAPPO with independent actor policies was removed due to a scalability issue on (b) with 1000 agents.

We observe that **IPPO** is the best-performing algorithm along with **InforMARL**. Conversely, standard implementations of **MAPPO** struggle to converge to an optimal policy. We observe that sharing actor-policy across agents in **MAPPO** performs better than having independent actor networks for each agent. This can be explained by the higher non-stationarity between agents with independent policies. However, policy sharing does not allow here to take into account the specificity of each agent's dependencies within the network because the inter-dependency is implicit in this problem. Hence, **MAPPO** with shared actor-policy converges towards a slightly sub-optimal policy while **InforMARL** can tackle the issue by using the graph structure of the problem.

Overall, the **IPPO** algorithm is surprisingly effective without any additional information or centralized component and aggregator. While slightly slower to converge, it reaches the best joint policy on every instance of the problem tested and scales very well up to $10,000$ agents (see Figure 8). **InforMARL** also scales very well but requires the communication of additional information to the agents and a centralized critic network with full knowledge of the state (or joint-observation). The standard **MAPPO** implementation remains largely suboptimal and does not scale when maintaining independent actor-policy networks. Policy sharing here helps with scalability and in converging towards a better joint policy.

## C.4   Open-source benchmark experiments

The code to reproduce all the experiments is available at `https://github.com/yojul/cognac-benchmark-example`. It contains all the training scripts for various algorithms, as well as the adjacency matrix of networks used to instantiate the environments.

## D   Training Settings for Benchmark Experiments

### D.1   Graph Structure

For simplicity, the network structure studied in the experiments for **BinaryConsensus**, **SysAdmin**, and **Multi-Commodity Flow** is defined through an adjacency matrix in the form of a band matrix, defining a directed graph where the number of outgoing edges for each node is fixed. The weights on the edges are drawn from a uniform distribution $w_i \in (0.25, 1)$. All adjacency matrices used in the experiments can be found along with the training scripts as `.npy` NumPy array files at `https://github.com/yojul/cognac-benchmark-example/tree/main/algos/env_assets`. For example, the network for $N = 10$ agents has the following adjacency matrix (values rounded to two decimals):

Table 6: Comparison of hyperparameters used for MAPPO and IPPO training.

| Hyperparameter | MAPPO Value | IPPO Value | Tested Range | Description |
|---|---|---|---|---|
| Learning rate ($\alpha$) | $3 \times 10^{-4}$ | $5 \times 10^{-4}$ | [1e-4, 1e-3] | Learning rate of the optimizer |
| Discount factor ($\gamma$) | 0.99 | 0.99 | – | Discount factor |
| GAE $\lambda$ | 0.90 | 0.95 | {0.9, 0.99} | $\lambda$ for Generalized Advantage Estimation |
| Number of minibatches | 1 | 4 | [1, 8] | Number of mini-batches per update |
| Update epochs ($K$) | 3 | 4 | [2, 10] | Number of policy update epochs |
| Clipping coefficient | 0.1 | 0.2 | [0.1, 0.5] | Surrogate clipping coefficient |
| Entropy coefficient | 0.01 | 0.02 | [0.01, 0.1] | Entropy regularization coefficient |
| Value loss coefficient | 0.5 | 0.5 | – | Value function loss coefficient |
| Max gradient norm | 0.5 | 0.5 | [0.5, 1.0] | Maximum gradient clipping norm |
| Rollout buffer size | 16 episodes | 10 episodes | [1, 32] | Rollout length (in episodes) |

$$\begin{bmatrix}
0.00 & 0.71 & 0.35 & 0.29 & 0.00 & 0.00 & 0.00 & 0.00 & 0.00 & 0.00 \\
0.30 & 0.00 & 0.78 & 0.98 & 0.51 & 0.00 & 0.00 & 0.00 & 0.00 & 0.00 \\
0.33 & 0.58 & 0.00 & 0.99 & 0.74 & 0.74 & 0.00 & 0.00 & 0.00 & 0.00 \\
0.98 & 0.89 & 0.81 & 0.00 & 0.28 & 0.64 & 0.74 & 0.00 & 0.00 & 0.00 \\
0.00 & 0.50 & 0.85 & 0.73 & 0.00 & 0.89 & 0.97 & 0.25 & 0.00 & 0.00 \\
0.00 & 0.00 & 0.76 & 0.36 & 0.61 & 0.00 & 0.55 & 0.51 & 0.73 & 0.00 \\
0.00 & 0.00 & 0.00 & 0.43 & 0.46 & 0.59 & 0.00 & 0.76 & 0.37 & 0.89 \\
0.00 & 0.00 & 0.00 & 0.00 & 0.38 & 0.89 & 0.90 & 0.00 & 0.71 & 0.95 \\
0.00 & 0.00 & 0.00 & 0.00 & 0.00 & 0.33 & 0.64 & 0.44 & 0.00 & 0.41 \\
0.00 & 0.00 & 0.00 & 0.00 & 0.00 & 0.00 & 0.83 & 0.98 & 0.65 & 0.00
\end{bmatrix}$$

**Note**   This setup can be easily extended to undirected settings by symmetrizing the matrix—i.e., by adding non-zero values to ensure that each node is undirectly connected to some of its neighbors.

The interpretation of adjacency matrix weights varies across environments. In **BinaryConsensus** and **SysAdmin**, the weights represent the probability that one agent influences another. In **BinaryConsensus**, an agent is influenced by its neighbors with a probability equal to the corresponding edge weight. Neighbors with incoming edges can affect the agent's state update, making some agents more influential than others, depending on their outgoing edge weights. In **SysAdmin**, agents in a *faulty* or *dead* state can spread their state to neighbors based on edge weight probabilities. As a result, nodes with high degrees and large edge weights play a more critical role in preventing the spread of failures.

In the **Multi-Commodity Flow Network**, edge weights represent flow costs. Minimizing flow along these edges directly impacts the agents' rewards. Importantly, in all environments, agents cannot observe the edge weights—they are part of the environment's internal dynamics.

We scaled down dependencies by a factor of 10 in our experiments with the Binary Consensus environment in order to help convergence for MAPPO and Q-Mix methods. This suggests that these kinds of methods struggle to capture strong dependency between agents using a centralized value function.

## D.2   Training Hyperparameters

For all experiments, we use a standard set of hyperparameters and perform minimal hyperparameter tuning beforehand. On small instances of problems, we expect independent algorithms to perform correctly. We tried to keep hyperparameters constant for all experiments for comparison purposes between problem structures and sizes. We describe the main set of parameters comparing IPPO and MAPPO in Table 6 and the set for IDQL in Table 7.

Table 7: Hyperparameters for Independent Q-Learning Experiments

| Parameter | Value / Description |
|---|---|
| **Training** | |
| Total timesteps | 5,000,000 |
| Learning starts | 10,000 steps |
| Training frequency | Every 5 steps |
| Optimizer | Adam |
| Learning rate | $5 \times 10^{-4}$ |
| **Exploration Schedule** | |
| Starting $\epsilon$ | 1.0 |
| Final $\epsilon$ | 0.005 |
| Exploration fraction | 0.1 of total timesteps |
| Exploration decay | Linear schedule |
| **Replay Buffer** | |
| Buffer size | 10,000 |
| Batch size | 256 |
| **Q-Learning Parameters** | |
| Discount factor ($\gamma$) | 0.95 |
| Soft update coefficient ($\tau$) | 0.5 |
| Target network update frequency | Every 100 steps |
| **Neural Network Architecture** | |
| Hidden layers | Two layers of 120 and 84 units. |
| Activation function | ReLU |
| Output | Q-values for each discrete action |

All experiments have been run on an Apple M3 Pro laptop using a single thread, i.e., each agent's neural network policy is called and trained sequentially. In all our experiments with up to 1000 agents, the time required to train working policies remains reasonable, from a few minutes to a few hours on a laptop.

