# OpenReview forum: "COGNAC: Cooperative Graph-based Networked Agent Challenges for Multi-Agent Reinforcement Learning"
_NeurIPS.cc/2025/Datasets_and_Benchmarks_Track — NeurIPS 2025 Datasets and Benchmarks Track poster_

### Official Review · Reviewer_dpix · 2025-06-26

**Rating:** 4
**Confidence:** 3

**Summary:**

The manuscript presents a new benchmark environment for multi-agent reinforcement learning (MARL), named COGNAC, specifically designed for fully cooperative multi-agent tasks with underlying graph structures. Targeting the widely encountered networked control problems in real-world scenarios, this work provides a flexible and scalable platform. The authors highlight the importance of decentralized and distributed learning, and support their claims by demonstrating the competitiveness of independent learning algorithms through empirical evaluations.

**Additional Feedback:**

See limitations weakness above. It is recommended that the authors systematically review and compare existing MARL benchmark environments and application cases to clearly define COGNAC’s unique contributions and scope of applicability, thereby further strengthening the paper’s positioning and impact.

**Dataset Code Accessibility:**

Yes

**Dataset Code Comments:**

The dataset and code are publicly available at [https://github.com/yojul/cognac], with accompanying documentation and metadata. The authors provide a stated motivation for the data collection, and the code includes instructions and dependency information that support replication. While some details could potentially be expanded, the provided resources are sufficient to enable reproducibility and further exploration.

**Ethical Comments:**

No.

**Ethical Considerations:**

No, there are no or only very minor ethics concerns

**Final Justification:**

Thank you for the author's response—my concerns have largely been addressed. I now understand the significance of the proposed benchmark in simulating large-scale agents, and therefore, I have decided to raise my rating to 4.

**Limitations Weaknesses:**

1. The discussion of prior work is somewhat limited. For example, the manuscript "Efficient and scalable reinforcement learning for large-scale network control", published in Nature Machine Intelligence, presents research along a similar line. It introduces the concept of networked MDPs, modeling a system with n agents as a graph, and systematically evaluates MARL algorithms across several real-world or high-fidelity simulated domains, including traffic, power systems, and healthcare. In comparison, although the proposed COGNAC environment appears to offer generality and scalability, it remains unclear whether its task complexity and realism are sufficient to capture the challenges encountered in actual systems. Furthermore, the manuscript does not clearly delineate the similarities and differences between COGNAC and existing benchmarks in terms of environment design and task difficulty. This lack of comparison may weaken the manuscript’s claims of being the "first" of its kind or serving as a "bridge" between MARL research and practical networked control applications.

2. The rationale for introducing graph structures as a central challenge in the environment is not sufficiently substantiated. The manuscript does not clearly articulate what specific difficulties graph-based formulations introduce for current MARL algorithms. For example, it remains uncertain whether even relatively simple graph topologies present challenges that existing methods are fundamentally unable to address. Without a more detailed analysis—either theoretical or empirical—demonstrating how and why graph structures complicate learning and coordination, the necessity of this design choice remains open to question. Clarifying this point would strengthen the case for the benchmark’s relevance and distinctiveness.

3. An important consideration in the construction of benchmark datasets is the trade-off between simplicity, flexibility, and realism. In many cases, fidelity to real-world conditions may be more critical than ease of use or adaptability. The current design of the environment appears to prioritize modularity and extensibility, but it is not evident whether this comes at the expense of faithfully capturing the complexity of real-world networked systems. A more thorough discussion of this design trade-off, including the authors’ justification for their choices, would be valuable for assessing the benchmark's practical applicability and relevance.

4. The organization and logical flow of the manuscript need improvement. For example, in the Introduction section, lines 21–22, 45–46, and 61–62 repeatedly emphasize the complexity of systems, with overlapping contextual content. Further refinement and consolidation are required.

**Strengths Contributions:**

1. The overall structure of the manuscript is clear and well-organized.

2. The task details are thoroughly described.

3. The initial release includes four distinct problem settings.

---

> ### Author Rebuttal · Authors · 2025-07-31
>
> We thank you for taking the time to review our paper and give us constructive feedback with an additional reference that we will add to our current review of existing work regarding realistic network control.
>
> To clarify our statement and contribution regarding the development of the COGNAC open-source package:
>
> - In COGNAC, we address the theoretically highly difficult problem of **finding optimal policies for Dec-POMDPs**. There are currently very few theoretical results and efficient methods to solve it. The main open challenge is to _identify special structures that make networked control problems theoretically solvable._ Thus, our package aims at providing a testbed platform to researchers from the **theoretical MARL** community.
> - Having flexible and transparent network dynamics and structures allows us to **isolate challenges** and facilitate the **identification of particular structures** and interesting properties to derive theoretical results, especially convergence guarantees for RL methods.
> Some examples of properties and studied challenges:
>     * **Network constraints**, such as edge constraints: implemented with edge capacity in the multi-commodity flow problem.
>     * **Cycles**: Recent results showed that Independent Q-Learning iterates have convergence guarantees in Directed Acyclic Graph structures. COGNAC can be used to empirically verify these results and be a powerful tool to identify potential extensions to derive new results.
>     * **Density**: We provide an experiment on the influence of network density on the learning process in Appendix B2 (Figure 4).
> - Therefore, from our perspective, this benchmark package mainly targets the theoretical MARL community and is thus complementary to more realistic benchmark environments (like the one presented in the paper you referenced).
> - Finally, from a purely technical perspective, we developed the package with full integration into modern RL research tools and want to engage the community:
>     * Environments inherit from _PettingZoo_ and _Gymnasium_, like other high-standard benchmark environments (e.g., MPE, Atari, SMAC, etc...).
>     * The package is deployed on the `Pypi` index, making it very easy to install through a simple `pip install cognac`.
>     * It comes with full technical documentation generated through _Sphinx_ and deployed on _ReadTheDocs_, which also includes tutorials, exhaustive descriptions of environments, and illustrations.
>
> ---
> Here are some elements to answer specific points of your feedback:
>
> > The manuscript does not clearly delineate the similarities and differences between COGNAC and existing benchmarks in terms of environment design and task difficulty. This lack of comparison may weaken the manuscript’s claims of being the "first" of its kind or serving as a "bridge" between MARL research and practical networked control applications.
>
> The development of COGNAC was initially motivated by the need to switch from realistic simulators (especially power grids and wind farms) to simpler problems with transparent, flexible, and controllable dynamics and structure.
>
> In our review of existing work, for clarity purposes, we chose to only include the most popular **benchmark environment packages**, which fall into two main categories:
> - toy problem benchmarks (i.e., relatively simple and usually supporting theoretical work)
> - reference realistic network control simulators for several problems directly linked to our area of ongoing research.
>
> We agree that a more in-depth and larger comparison with existing work is important, so we propose to include the table below. The table highlights how COGNAC fills a gap as a suite of benchmark environments for **network control** tasks within the category of **_toy problem benchmarks_**, allowing for scaling problems **up to 10k agents** (max. instance size tested) with **partial observability** and **high modularity** for problem definition.
>
>
> | **Benchmark**                              | **Task type(s)**             | **Max # agents** | **Modular** | **Partial Obs.** |
> |-------------------------------------------|------------------------------|------------------|-------------|------------------|
> | *Toy problems*                                   |                              |                  |             |                  |
> | Multi-Particle Env.                       | Navigation / Communication   | 6                | ✘           | ✔                |
> | Google Football                           | Navigation / Control         | 22               | ✘           | ✔                |
> | LB-Foraging                               | Navigation                   | ~10              | ✔           | ✘                |
> | SMACv2                                    | Navigation / Control         | ~100             | ✘           | ✔                |
> | POGEMA                                    | Navigation                   | ~100             | ✔           | ✔                |
> | MeltingPot                                | Various                      | ~10              | ✘           | ✔                |
> | Multi-Agent Atari                         | Various                      | 4                | ✘           | ✘                |
> | SISL                                      | Navigation / Control         | ~8               | ✘           | ✔                |
> | Overcooked                                | Navigation / Control         | 2                | ✘           | ✘                |
> | **COGNAC**                                | **Network Control**          | **~10,000**      | **✔**       | **✔**            |
> | *Realistic environments*  |                              |                  |             |                  |
> | Grid2Op                                | Network Control              | ~100             | ✘           | ✘                |
> | WFCRL                                  | Network Control              | ~30              | ✔           | ✔                |
> | SUMO-RL                                | Network Control              | ~20              | ✘           | ✔                |
> ---
>
> In this context, we argue that COGNAC fills a gap while remaining fully complementary with pre-existing realistic simulators as well as classic MARL benchmark environments.
>
> ---
> > Without a more detailed analysis—either theoretical or empirical—demonstrating how and why graph structures complicate learning and coordination, the necessity of this design choice remains open to question.
>
> The problems defined in COGNAC fall in the category of Decentralized Partially Observable Markov Decision Processes (Dec-POMDPs), which are known to be highly difficult, i.e., **N-EXP Complete** (Oliehoek, 2016). Therefore, the underlying objective of COGNAC’s development is to support research for theoretical results that leverage the network structure of a problem to tackle the general complexity of finding an optimal policy.
>
> **Results and analysis**
> - Appendix B2, Figure 4: We show that increasing the density of the graph, i.e., increasing dependency between agents, generally makes learning more difficult, as the return tends to drop with a fixed learning process.
> - We performed recent experiments to compare convergence speed on different graph structures: Directed Graph, Undirected Graph, and Directed Acyclic Graph. These experiments support recent theoretical and empirical results on the convergence of Independent Q-Learning iterates in Directed Acyclic Graphs (Bizon-Monroc, 2024). Tested graph structures are included in the assets folder of the benchmark experiments code repository. Moreover, utility functions to generate specific graph structures are directly part of the package.
>
> ---
> > A more thorough discussion of this design trade-off, including the authors’ justification for their choices, would be valuable for assessing the benchmark's practical applicability and relevance.
>
> COGNAC was designed to address the lack of experimental platforms **supporting theoretical results** for MARL methods on network systems. The development of COGNAC was initially motivated by the need to switch from realistic simulators (especially power grids and wind farms) to simpler problems with transparent, flexible, and controllable dynamics and structures.
>
> Therefore, we believe in the choice of implementing toy environments with simple dynamics and high flexibility over size and structure, rather than realistic problems and dynamics. At the same time, we fully agree on the concurrent need for benchmark environments that fully capture the complexity of network systems, such as the environment proposed in your suggested reference.
>
> ---
> > The organization and logical flow of the manuscript need improvement. For example, in the Introduction section, lines 21–22, 45–46, and 61–62 repeatedly emphasize the complexity of systems, with overlapping contextual content. Further refinement and consolidation are required.
>
> Thank you for this comment. Some repetitions like the ones highlighted here will be refined.
> ___
> ### Reference
> 1. Bizon-Monroc, C., Bušić, A., Dubuc, D., & Zhu, J. (2024). Wind farm control with cooperative multi-agent reinforcement learning. In ICML 2024 Workshop on Aligning Reinforcement Learning Experimentalists and Theorists (ARLET 2024).
> 2. Oliehoek, F. A., & Amato, C. (2016). A concise introduction to decentralized POMDPs (Vol. 1). Cham, Switzerland: Springer International Publishing.

---

> > ### Comment · Reviewer_dpix · 2025-08-06
> > **I have decided to raise my rating to 4.**
> >
> > Thank you for the author's response—my concerns have largely been addressed. I now understand the significance of the proposed benchmark in simulating large-scale agents, and therefore, I have decided to raise my rating to 4.

---

### Official Review · Reviewer_FhwH · 2025-07-02

**Rating:** 4
**Confidence:** 2

**Summary:**

The paper has made two key contributions to benchmark of graph-based MARL.

(1) The authors develop a RL benchmark for cooperative MARL with graph structure.

(2) The authors provide open-source implementations of several graph-based MARL algorithms.

**Dataset Code Accessibility:**

Yes

**Dataset Code Comments:**

The data and code are made publicly available. The user guideline is also provided. I didn't check the implementation and the reproductivity.

**Ethical Considerations:**

No, there are no or only very minor ethics concerns

**Final Justification:**

Viewing the value of the networked MARL benchmark environment, I would like to increase the score by 1.

**Limitations Weaknesses:**

(1) Section 2: It is not very helpful to list details of available environments. I would recommend highlighting state-of-the-art algorithms in each environment and their challenges.

(2) Section 3: Do experimental results support theoretical results for each method? This is not analyzed yet.

(3) Section 3: Do experimental results explain influence of different graph topologies and scalability in terms of the graph size? These are also not analyzed yet.

**Strengths Contributions:**

(1) The studied graph-based MARL problems is widely used in network systems and so it is a real problem. Since practical performance of theoretical MARL algorithms is not well evaluated in literature, the paper fills in this gap.

(2) The proposed benchmark can handle different networks with various sizes and properties.

---

> ### Author Rebuttal · Authors · 2025-07-31
>
> Thank you for taking the time to review our paper and for your feedback. However, we have to respectfully disagree with most of the comments you make on our work, and we think that part of our contribution to MARL research is misunderstood here.
>
> ## Some clarification statements
> 1. COGNAC package provides MARL researchers a **suite of benchmark environments** to evaluate and test algorithms on network control problems. Please see the table comparing our package to pre-existing work provided to Reviewer 3. It highlights the unique positioning of COGNAC among popular benchmark environment packages for MARL.
> 2. The general underlying problem of MARL, i.e., Dec-POMDPs, has **no general grounded theoretical results** as it is known to be N-EXP complete in the general case (Oliehoek, 2016).  As we tackle highly difficult problems here with Dec-POMDPs and introduce new benchmark problems, there are no known analytical solutions for the given problems, nor grounded convergence guarantees for most of the MARL algorithms tested, especially when they involve deep learning-based policies.
> Thus, no analysis comparing empirical results to theory can be done here.
> 3. Our package provides a platform to address a **key open challenge** in this field: _Identify special structures that make problems theoretically solvable._
>     * Example of recent results: Convergence of Independent Q-Learning Iterates for problems with Directed Acyclic Graph structures (Bizon-Monroc, 2024).
>     * To this extent, we propose several experiments regarding RL on graph-structured problems:
>         - Centralized vs Fully Decentralized MARL paradigms (Results, Figure 2)
>         - Graph density influence (Appendix B2, Figure 4)
>         - Graph size (Appendix B3, Figure 5)
>         - Graph macro-structure: acyclic vs cyclic vs dense (experiments in code repository)
>
> Overall, the lack of theoretical results in the literature is a primary concern motivating the development of COGNAC. We hope that our benchmark will contribute to the development of future theoretical work in the field of MARL.
>
> ## Answers to specific comments
>
> > Section 2: It is not very helpful to list details of available environments. I would recommend highlighting state-of-the-art algorithms in each environment and their challenges.
>
> We disagree with this statement, as we believe that the developed benchmark environments constitute the **main contribution** of this paper.
>
> We intentionally provided full descriptions of each COGNAC task because two of them (Binary Consensus, Multi‑commodity Flow) are brand‑new graph‑structured Dec‑POMDP formulations of classical problems (that were not originally stated as an MDP), and with no prior implementations. This also implies that currently there are no state-of-the-art baselines specific to these problems. The Firefighting and SysAdmin tasks are standard benchmark problems from Dec-POMDP literature but lack full standard descriptions and open-source implementations. To keep generality from the MARL perspective, we chose to evaluate standard, widely used model-free algorithms on all environments.
>
> Hereafter is the list of baselines for each environment (these baselines often assume either full observability or some knowledge of the system dynamics):
> - **Firefighting Graph:** This problem can be represented as a Directed Bayesian Network and has a weakly coupled structure. In the 1D case, one can decompose the value function into a sum of local value functions (assuming a factored reward function). Thus, it is in theory possible to solve the problem exactly (at least in the 1D case). However, it remains tractable only for small instances of the problem. One can still leverage value decomposition to get approximate solutions through planning methods as proposed in (Oliehoek, 2013).
> - **SysAdmin Network:** Initially introduced in (Guestrin, 2002), it leverages the structure of the problem as a Networked MDP to find approximate solutions through Max-norm projection. In this initial paper, the problem is solved with a centralized planning method assuming knowledge of the dynamics and small, tractable instances of the problem (~10 agents). More recent work uses Monte Carlo Tree Search (MCTS) methods (Bianchi, 2024) or model-based methods such as Cooperative Prioritized Sweeping (Bargiacchi, 2021), also tested on Firefighting Graph.
> - **Binary Consensus:** This is a brand-new problem inspired by the voter model. There are algorithms that can solve it assuming knowledge of the dynamics. Therefore, there is currently no baseline algorithm regarding a similar RL-based (or MDP) formulation of the problem.
> - **Multi-commodity Flow Problem:** This is a classic control problem; however, to the best of our knowledge, this is the first implementation of this particular problem in a multi-agent setting where the objective is to control the system from the perspective of network nodes as agents. Thus, we could not identify any particular baselines for this setting, although it has some similarities with multi-agent traffic control or topology management in power grids. Generally, the multi-commodity flow problem with integer flows is NP-Complete, but there are $(1+\epsilon)$ approximation algorithms to solve it.
>
> Updating to better heuristic baselines for each problem is a work in progress. In addition, we are planning to implement baseline agents directly in the package for comparison and imitation learning purposes.
>
> ---
> > Section 3: Do experimental results support theoretical results for each method? This is not analyzed yet.
>
> Regarding environments, Dec-POMDP problems are known to be N-EXP complete in the general case (Oliehoek, 2016). Thus, there are no analytical solutions to the proposed environments, which also motivates the need for RL methods in such systems.
>
> Regarding MARL algorithms, there are very few theoretical results. In particular, PPO and its associated multi-agent adaptations (MAPPO and IPPO) have no theoretical convergence guarantee. While demonstrating very good results on many problems, including COGNAC’s environments, motivations for implementing these algorithms are purely empirical. As mentioned, there are a few results for the convergence of Independent Q-Learning; we cite the guarantee obtained in the particular case of Directed Acyclic Graph structures under multi-scale learning rates (Bizon-Monroc, 2024).
>
> Therefore, there are no theoretical results that can be analyzed in relation to experiments here. However, we think that COGNAC is a valuable asset to identify special structures in network problems to derive new theoretical results.
>
> ---
> > Section 3: Do experimental results explain the influence of different graph topologies and scalability in terms of the graph size? These are also not analyzed yet.
>
> We proposed several experimental results regarding the graph size and topology influence, one in the main paper and two in the supplementary material:
>
> ### In the main text of the paper:
> - (3. Results) **Figure 2:** The comparison between centralized PPO and IPPO demonstrates the limits of the centralized method when scaling one of the environments from 10 to 100 agents. This highlights the clear advantage of decentralized RL algorithms over a naive centralized approach (i.e., treating the environment as a single-agent problem).
> - (Appendix B2) **Figure 4:** We study the influence of graph density on the learning process. Specifically, we observe how the final performance of the learned policy degrades as the density of the graph increases with a fixed learning process.
> - (Appendix B3) **Figure 5:** We scale up the SysAdmin Network to 10k agents, highlighting the capability of our package to study very large multi-agent systems. As a comparison, the maximum number of agents in standard MARL benchmark environments such as StarCraft is around 50 agents.
>
> ### Current work:
> We recently added results highlighting the convergence speed of standard MARL algorithms when learning on DAG-structured instances compared to more standard network structures with cycles (Directed, Undirected, Dense networks). The graph structures supporting the experiments are available in the experiment repository.
>
> ---
> ### Reference
> 1. Bizon-Monroc, C., Bušić, A., Dubuc, D., & Zhu, J. (2024, July). Wind farm control with cooperative multi-agent reinforcement learning. In ICML 2024 Workshop on Aligning Reinforcement Learning Experimentalists and Theorists (ARLET 2024).
> 2. Oliehoek, F. A., & Amato, C. (2016). A concise introduction to decentralized POMDPs (Vol. 1). Cham, Switzerland: Springer International Publishing.
> 3. Oliehoek, F. A., Whiteson, S., & Spaan, M. T. (2013). Approximate solutions for factored Dec-POMDPs with many agents.
> 4. Guestrin, C., Koller, D., & Parr, R. (2001, August). Max-norm projections for factored MDPs. In IJCAI (Vol. 1, pp. 673-682).
> 5. Bianchi, F., Castellini, A., Farinelli, A., Marzari, L., Meli, D., Trotti, F., & Veronese, C. (2024). Developing safe and explainable autonomous agents: from simulation to the real world. In CEUR WORKSHOP PROCEEDINGS (pp. 89-94).
> 6. Bargiacchi, E., Verstraeten, T., & Roijers, D. M. (2021, May). Cooperative Prioritized Sweeping. In AAMAS (pp. 160-168).

---

### Official Review · Reviewer_fQXD · 2025-07-02

**Rating:** 5
**Confidence:** 5

**Summary:**

The paper introduces COGNAC, a suite of benchmark environments for multi-agent reinforcement learning (MARL) specifically designed for fully cooperative tasks with inherent graph structures. The authors, aim to bridge the gap between theoretical research in network control and practical MARL applications by providing a flexible and scalable platform.

**Additional Feedback:**

- Given the acknowledged difficulty in computing theoretically optimal policies, are there any approximations or bounds that could be used as more rigorous baselines, especially for the simpler problem instances?
- The paper mentions that GNNs could be promising for network and graph-based problems. Have the authors considered integrating GNN-based agents into COGNAC? I think including them in the benchmark would make it a stronger paper for the datasets and benchmark track.
- While IDQL and IPPO show promising results, MARL research has explored various coordination mechanisms. Do the authors plan to expand the benchmark examples to include more advanced multi-agent methods, such as those leveraging centralized critics (e.g., MAPPO) or explicit communication between agents, to further highlight the challenges and advantages of different paradigms within COGNAC?
- For environments like Binary Consensus and SysAdmin that are graph-agnostic, how do different graph structures quantitatively impact the performance of various MARL algorithms? Is it possible to include an analysis for this?
- It would be easy for the readers if an illustrative figure for the environments could be included in the paper.

**Dataset Code Accessibility:**

Yes

**Dataset Code Comments:**

More documentation about usage of environments is required (I think the authors are working on this)

**Ethical Considerations:**

No, there are no or only very minor ethics concerns

**Final Justification:**

Thanks for the reply! I would like to increase my score by 1 since the authors have provided enough information on the hyperparameter tuning. I would expect the authors to include this in the final camera ready paper including the future work/extensions to graph-based methods.

**Limitations Weaknesses:**

- The authors state they perform minimal hyperparameter tuning for the benchmark experiments, which might not showcase the full potential of the evaluated algorithms.
- While foundational, the initial benchmark only evaluates independent learning algorithms, and the authors themselves suggest other distributed methods, including Graph Neural Networks (GNNs), could yield better results. I would be curious to see how does adding “multi-agentness” (either through the RL update or through communication) in the underlying algorithms affect the performance. The authors state that the benchmark could be easily extended to MAPPO and other approaches using centralized critics. I would urge the authors to include experiments with MAPPO, InforMARL (graph-based extension of MAPPO), etc.
- Figure 2a: Centralized PPO is able to achieve a mean reward of ~400 at around 200k steps. Which makes me think if the the training was performed until an optimal policy was achieved for the binary consensus problem. Could you please explain the peak and the drop in the plot?

References:
1- MAPPO: Yu, C., Velu, A., Vinitsky, E., Gao, J., Wang, Y., Bayen, A., & Wu, Y. (2022). The surprising effectiveness of ppo in cooperative multi-agent games. Advances in neural information processing systems, 35, 24611-24624.
2- InforMARL: Nayak, S., Choi, K., Ding, W., Dolan, S., Gopalakrishnan, K., & Balakrishnan, H. (2023, July). Scalable multi-agent reinforcement learning through intelligent information aggregation. In International Conference on Machine Learning (pp. 25817-25833). PMLR.

**Strengths Contributions:**

- The paper clearly identifies and fills a need for benchmark environments specifically designed for fully cooperative multi-agent tasks with inherent graph structures. I believe that this set of standardized environments is a good addition for the research community but would like to see more benchmarking against other commonly used MARL methods.
- I appreciate that COGNAC is designed for adaptability, allowing users to customize environment parameters and scale problem sizes.
- Each of the four problems (Firefighting Graph, Binary Consensus, SysAdmin, Multi-commodity Flow) is thoroughly described, including details on their state space, action space, and objective functions. Although, I would appreciate if the authors could document the code repository (it states as documentation in progress when I check the repo)
- I like how Table 1 summarizes the differences in the features of the different environments. It would be good to have a similar table comparing COGNAC with other methods.

---

> ### Author Rebuttal · Authors · 2025-07-30
>
> Thank you very much for your time and your valuable feedback.
>
> > I would appreciate if the authors could document the code repository
>
> The Sphinx‐generated technical documentation was deployed on ReadTheDoc with the initial release. It was tagged _in progress_ because tutorials, exhaustive descriptions, and illustrations were still pending. It’s now complete.
>
> ---
> >The authors state they perform minimal hyperparameter tuning for the benchmark experiments, which might not showcase the full potential of the evaluated algorithms
>
> What we meant was that we used the standard hyperparameter sets from the literature without further fine‐tuning to ensure fair benchmarks. We found IPPO and IDQL largely insensitive to tuning. We will clarify this in the text and add a tuned‐parameters file to the experiments repo to illustrate each method’s best performance.
>
> ---
> > How does adding “multi-agentness” in the underlying algorithms affect the performance. The authors state that the benchmark could be easily extended to MAPPO and other approaches using centralized critics. I would urge the authors to include experiments with MAPPO, InforMARL, etc.
>
> We added Q-Mix and MAPPO to the benchmark repository. We observe that these methods are much more sensitive to hyperparameter tuning. In addition, neither Q-mix nor MAPPO were able to outperform IPPO on all tested instances and slightly outperform IDQL on some instances.
>
> As standard methods using central critic struggle to outperform IPPO, this suggests that more advanced coordination mechanisms may be needed, that better exploit the graph structure of the environments. Next step, as you suggested, is implementing InforMARL (in progress). Ultimately, we expect to ourperform IPPO.
>
> ---
> > Figure 2a: Centralized PPO is able to achieve a mean reward of ~400 at around 200k steps. [...] Could you please explain the peak and the drop in the plot?
>
> The peak and subsequent drop were caused by a flaw in the initial reward design (detailed below), which encouraged the centralized policy to extend episodes near consensus instead of reaching consensus quickly. This problem was not observed when doing IPPO because each agent only exploits its local reward signal. It is now solved and both converge to similar episode return, with different convergence speed (IPPO being faster).
>
> ## Reward Design
>
> _Note: For flexibility reason, one can design any reward function compatible with the environment inheriting from a `BaseReward` class. For each environment, we designed simple default rewards that comply with the objective of the problem (consensus, task completion, etc...)_
>
> ### Former Reward
> - **During an episode** - Each agent gets a local reward:
>   $$r_i(t)=\begin{cases}+1&\text{if agent }i\text{ agrees with majority } \\\\ -1 & \text{otherwise} \end{cases}$$
> - **At episode end**: Agents receive a large terminal reward for reaching the consensus as fast as possible. If the horizon is reached without a consensus they get a large negative reward weighted by how close the system was to the consensus. Formal detailed description can be found within reward technical documentation.
>
> **Issue:** This rewards each agent locally with $+1$ for agreeing with majority, leading to the global reward (as the sum of local rewards) being positive at each step when **being close to consensus but not reaching it**. This incentivizes the centralized policy to make episode last as long as possible at the expense of high variance in the final return (as agent might not reach the consensus before the end). Ultimately, it converges to a policy with less variance that is also suboptimal.
>
> ### New reward
> **During an episode** - Each agent get a local reward:
>   $$r_i(t)=\begin{cases}0&\text{if agent }i\text{ agrees with majority} \\\\ -1&\text{otherwise}\end{cases}$$
>
> ---
> > Given the acknowledged difficulty in computing theoretically optimal policies, are there any approximations or bounds that could be used as more rigorous baselines, especially for the simpler problem instances?
>
> We are currently working on implementing improved baselines agents for each problem directly into the package. This will be useful for evaluation and training purpose (e.g. imitation learning).
>
> - **FirefightingGraph**: In the 1D case, one can decompose the value function into a sum of local values function (assuming a factored reward function). Thus, it is in theory possible to solve the problem exactly (at least in the 1D case). However, it remains tractable only for small instance of the problem. One can still leverage the value decomposition to get approximate solutions through planning methods as proposed in (Oliehoek, 2013).
> - **SysAdmin Network**: the original paper leverages the structure of the problem as a Networked MDP to find approximate solution through Max-norm projection (Guestrin, 2001). This planning method assumed full observability, knowledge of the dynamics and centralized control of the system. It is still intractable when scaling to 1000 agents.
> - **BinaryConsensus**: This is a new MDP problem, inspired by the voters model. So, there is no heuristics yet.
>     * Optimal policy with full-state knowledge: Assuming that all the weights on edges (influence between agents) are $w_{ij}<1,\forall(i,j)$, then for any initial state, there exist an optimal action such that the probability to reach the consensus is non-zero.
>     Note: For some strongly coupled structure with deterministic edges, there exists initial states where the consensus is unreachable within the time horizon.
>     * We are currently deriving a more advanced heuristic baseline algorithm for reaching the consensus using only the local observation and reward signal for each agent.
> - **Multi-Commodity Flow**: The current baseline is a rounded solution to the Linear Program that maximizes the total flow utility across the network. More advanced $(1+\epsilon)$ approximate methods improve the rounding to more optimal solution. These solutions solve the problem centrally observing the full network. Thus it is untractable for larger network and/or number of commodities.
>
> ---
> > Have the authors considered integrating GNN-based agents into COGNAC?
>
> Adding GNN-based methods is indeed relevant.
> - Algorithms: we are planning to implement methods such as InforMARL from the paper you referenced (thank you).
> - Graph-friendly features: We will release new features that allows to directly get graph observations for relevant environments. Currently, we use the `networkx` package dedicated to graph manipulations for internal operations, generation and plotting utilities. We also plan on adding native `torch geometric` compatibility.
>
> ---
> > Do the authors plan to expand the benchmark examples to include more advanced multi-agent methods, such as those leveraging centralized critics (e.g., MAPPO) or explicit communication between agents, to further highlight the challenges and advantages of different paradigms within COGNAC?
>
> As our main objective lies in studying large scale system, starting from independent learning was a deliberate choice and we aim at frugal communication mechanisms. While proven to be very efficient on many problems, we observe that tested Centralized Training Decentralized Execution (CTDE) methods underperform on COGNAC’s environments. Firstly, we observe that these methods are sensitive to hyperparameter tuning and hence, often yield suboptimal policies. Secondly, in systems with a large number of agents, it is likely that the size of the state space becomes intractable for the centralized critic (e.g. with 10k agents, see Appendix B3. Figure 5).
>
> ---
> > For environments like Binary Consensus and SysAdmin that are graph-agnostic, how do different graph structures quantitatively impact the performance of various MARL algorithms?
>
> We worked on some experiments in this regard:
> - In Appendix B2 Figure 4: It shows how density of graph (i.e. how much agents influence each other in the system) impacts learning. Indeed, increasing density correlates with decreasing final performance with a fixed learning process. This might have its place in the main text of the paper.
> - We performed recent experiments to compare convergence speed on different graph structures: Directed Graph, Undirected Graph, Directed Acyclic Graph. These experiments support recent theoretical and empirical results on the convergence of Independent Q-Learning Iterates in Directed Acyclic Graph (Bizon-Monroc, 2024). Tested graph structures are included in the `env_assets` folder of the benchmark experiments code repository. Moreover, utility functions to generate graph structures are directly part of the package.
> - The Multi-Commodity Flow problem is now also graph-agnostic to enable experiments on properties and structures.
>
> ---
> > It would be easy for the readers if an illustrative figure for the environments could be included in the paper.
>
> We added illustration figures for each environment in the documentation. They will be incuded in the paper as well.
>
> ---
> > It would be good to have a similar table comparing COGNAC with other methods.
>
> Thank you for this suggestion, third reviewer raises a similar comment. We wrote the table, please see rebuttal for reviewer 3.
> Indeed, it gives a clearer overview of available open-source benchmark environments for both toy problems and realistic simulators dedicated to network control.
>
> ---
> ### Reference
> 1. Bizon-Monroc, C., Bušić, A., Dubuc, D., & Zhu, J. Wind farm control with cooperative multi-agent reinforcement learning. In ARLET 2024, ICML 2024 Workshop.
> 2. Oliehoek, F. A., & Amato, C. (2016). A concise introduction to decentralized POMDPs (Vol. 1). Springer International Publishing.
> 3. Oliehoek, F. A., Whiteson, S., & Spaan, M. T. (2013). Approximate solutions for factored Dec-POMDPs with many agents. AAMAS 2013.
> 4. Guestrin, C., Koller, D., & Parr, R. (2001). Max-norm projections for factored MDPs. IJCAI 2001.

---

### Note · Authors · 2025-08-13

We summarize here the outcome of the rebuttal and discussion phase, during which many comments have been addressed thanks to the insightful feedback from the reviewers.

1. **Two new algorithms** have been added to the benchmark: **Q-Mix** and **MAPPO**. This should support a larger evaluation for state-of-the-art MARL algorithms on COGNAC's environments.

2. **Hyperparameter exploration and tuning** has been performed to ensure that each methods is well exploited. This should support a more solid set of baselines results for each environmets.

3. A more indepth litterature review has been added together with a **table comparinge existing benchmark environments**. This includes additional references as suggested by the reviewers.

4. A flaw in the reward design of one problem has been identified and fixed. The results obtained when training a centralized policy on _Binary Consensus_ problem now converges towards an optimal policy as expected.

In addition, the main text will be refined following reviewers' suggestions and we will clarify all points raised regarding parameter tuning, results comparison and analysis, advanced baselines for analytical or approximate optimal policy. on each problem, and the main open challenge adressed. We will also try to add results for graph-based algorithms.

Regarding open-source availability, our package is now **fully deployed** on the `PyPI` index, and the code for both environments and benchmark experiments is up to date on GitHub, accompanied by full **user-friendly documentation**.

We thank the reviewers again for their valuable feedback and the constructive discussion phase.

---

### Decision · Program_Chairs · 2025-09-18

**Decision:**

Accept (poster)

**Comment:**

This paper introduces COGNAC, which provides cooperative multi-agent environments with inherent graph structures for distributed multi-agent reinforcement learning (MARL). It aims to address critical challenge regarding lack of standardized and to provide a testbed for this increasingly important research problem. Four environments were included in the initial release with enough descriptions on objectives and formulations

Concerns were raised on hyperparameter tuning, positioning of this benchmarking suite, as well as clarity on its key challenges, which were all clearly discussed and well resolved in the discussion phase. In the future revisions, it is suggested to incorporate these comments accordingly to further improve the clarity. Besides, it is also highly appreciated on the continuous efforts to deploy the suite, include new algorithms, and sanity check on the implementations.